# Using population selection and sequencing to characterize natural variation of starvation resistance in *Caenorhabditis elegans*

**Amy K Webster[1†], Rojin Chitrakar[1], Maya Powell[1‡], Jingxian Chen[1], Kinsey Fisher[1], Robyn E Tanny[2], Lewis Stevens[2§], Kathryn Evans[2], Angela Wei[1], Igor Antoshechkin[3], Erik C Andersen[2], L Ryan Baugh[1,4]***

[1]Department of Biology, Duke University, Durham, United States; [2]Department of Molecular Biosciences, Northwestern University, Evanston, United States; [3]Division of Biology, California Institute of Technology, Pasadena, United States; [4]Center for Genomic and Computational Biology, Duke University, Durham, United States

*For correspondence:
ryan.baugh@duke.edu

Present address: [†]Institute of Ecology and Evolution, University of Oregon, Eugene, United States; [‡]Environment, Ecology, and Energy Program, University of North Carolina, Chapel Hill, United States; [§]Tree of Life, Wellcome Sanger Institute, Cambridge, United States

**Abstract** Starvation resistance is important to disease and fitness, but the genetic basis of its natural variation is unknown. Uncovering the genetic basis of complex, quantitative traits such as starvation resistance is technically challenging. We developed a synthetic-population (re)sequencing approach using molecular inversion probes (MIP-seq) to measure relative fitness during and after larval starvation in *Caenorhabditis elegans*. We applied this competitive assay to 100 genetically diverse, sequenced, wild strains, revealing natural variation in starvation resistance. We confirmed that the most starvation-resistant strains survive and recover from starvation better than the most starvation-sensitive strains using standard assays. We performed genome-wide association (GWA) with the MIP-seq trait data and identified three quantitative trait loci (QTL) for starvation resistance, and we created near isogenic lines (NILs) to validate the effect of these QTL on the trait. These QTL contain numerous candidate genes including several members of the Insulin/EGF Receptor-L Domain (*irld*) family. We used genome editing to show that four different *irld* genes have modest effects on starvation resistance. Natural variants of *irld-39* and *irld-52* affect starvation resistance, and increased resistance of the *irld-39; irld-52* double mutant depends on *daf-16/FoxO*. DAF-16/FoxO is a widely conserved transcriptional effector of insulin/IGF signaling (IIS), and these results suggest that IRLD proteins modify IIS, although they may act through other mechanisms as well. This work demonstrates efficacy of using MIP-seq to dissect a complex trait and it suggests that *irld* genes are natural modifiers of starvation resistance in *C. elegans*.

## Editor's evaluation

The authors identify natural genetic variants in *C. elegans* that are associated with variation in starvation resistance. The authors focus on a gene family (irld's) that are thought to regulate insulin signaling. These studies are very interesting in that the approach for identifying natural gene variants is highly innovative and the work provides novel information about this family of genes.

## Introduction

Given tremendous sequencing capacity, digitization of phenotypes by counting DNA molecules in mixed-genotype populations can provide unprecedented sensitivity and precision.

Population-selection-and-sequencing approaches for genetic analysis were developed in bacteria and yeast, enabling large numbers of genetic perturbations to be assayed in parallel (*Han et al., 2010*; *Kwon et al., 2016*; *Nislow et al., 2016*), and CRISPR subsequently enabled related approaches in mammalian cells (*Gilbert et al., 2014*; *Koike-Yusa et al., 2014*; *Shalem et al., 2014*; *Wang et al., 2014*). With its small size, genetic toolkit, and genomic resources, the nematode *C. elegans* is an ideal animal model to develop selection-and-sequencing approaches to organismal phenotypes. Such approaches have been described for mapping causal loci from recombinants between a pair of divergent strains in *C. elegans* (*Mok et al., 2017*; *Burga et al., 2019*). We described a population-sequencing approach based on pooling many wild strains (*Webster et al., 2019*), but it lacked power since only very rare sequencing reads that include single-nucleotide variants (SNVs) unique to a strain in the pool informed inference of relative strain frequency. By capturing targeted sequences, MIP-seq enables extremely deep sequencing of polymorphic loci (*Cantsilieris et al., 2017*; *Mok et al., 2017*), but it has not been applied to populations of wild strains.

Enduring periods of starvation is a near-ubiquitous feature of animal life that affects survival, growth, and reproduction, making starvation resistance a fitness-proximal trait. Starvation resistance is also important to human health and disease, with direct relevance to diabetes, obesity, aging, and cancer. Despite its importance to understanding animal evolution and informing therapeutic strategies, however, the genetic basis of natural variation in starvation resistance is unclear. The nematode *C. elegans* is frequently starved in the wild and has robust starvation responses (*Schulenburg and Félix, 2017*; *Baugh and Hu, 2020*). Larvae that hatch in the absence of food arrest development in the first larval stage (L1 arrest) and can survive starvation for weeks (*Baugh, 2013*). In addition to causing mortality, extended starvation reduces growth and reproductive success upon feeding (*Jobson et al., 2015*; *Jordan et al., 2019*), and these effects can be uncoupled by genotype or condition (*Roux et al., 2016*; *Kaplan et al., 2018*; *Chen et al., 2022* and reviewed in *Baugh and Hu, 2020*). Starvation resistance therefore integrates survival, growth rate, and reproductive success, and different genes and conditions can affect these phenotypes independently. Insulin/IGF signaling (IIS) is a critical regulator of L1 arrest (*Baugh and Sternberg, 2006*). There is a single known insulin/IGF-like receptor in *C. elegans*, DAF-2/InsR, which signals through a conserved phosphatidylinositol 3-kinase (PI3K) signaling pathway to antagonize the transcription factor DAF-16/FoxO (*Lin et al., 1997*; *Ogg et al., 1997*). When IIS is reduced, such as during starvation, DAF-16 moves to the nucleus and regulates transcription (*Henderson and Johnson, 2001*; *Lee et al., 2001*; *Lin et al., 2001*), promoting starvation resistance (*Muñoz and Riddle, 2003*; *Baugh and Sternberg, 2006*; *Hibshman et al., 2017*). (*Roux et al., 2016*; *Kaplan et al., 2018*; *Baugh and Hu, 2020*; *Chen et al., 2022*).

Here, we describe the development of MIP-seq for statistical genetic analysis of complex traits in *C. elegans*. We used MIP-seq to analyze starvation resistance in a pool of genetically diverse wild strains, identifying relatively starvation-resistant and sensitive strains. We identified and validated three QTL that affect starvation resistance and contain numerous candidate variants. Our results suggest that multiple members of the *irld* gene family affect aspects of starvation resistance, and they suggest they do so at least in part by modifying IIS.

## Results

### Sensitive and precise measurement of strain frequency in pooled culture using MIP-seq

We selected 103 genetically diverse, wild *C. elegans* strains from around the world including the laboratory reference N2 to test MIP-seq, ultimately phenotyping 100 for starvation resistance (*Figure 1A–B*). MIPs are designed to capture a specific region of the genome for targeted multiplex sequencing (*Figure 1C*). We designed MIPs to target a region containing a SNV unique to each of 103 strains. Thus, the relative frequency of each strain in a pool can be determined by the SNV frequency. We designed four such MIPs per strain to provide redundancy and increase precision. To pilot MIP-seq, we prepared sequencing libraries from an equimolar mix of genomic DNA from each of 103 strains. We determined the frequency of strain-specific reads for each MIP, and we censored probes that produced frequencies substantially different than the expected value of approximately 0.01 (*Figure 1D*; criteria in Materials and methods), leaving three or four reliable probes for 85% of strains and at least one MIP for 100 strains (*Figure 1E*, *Supplementary file 1*), which were used

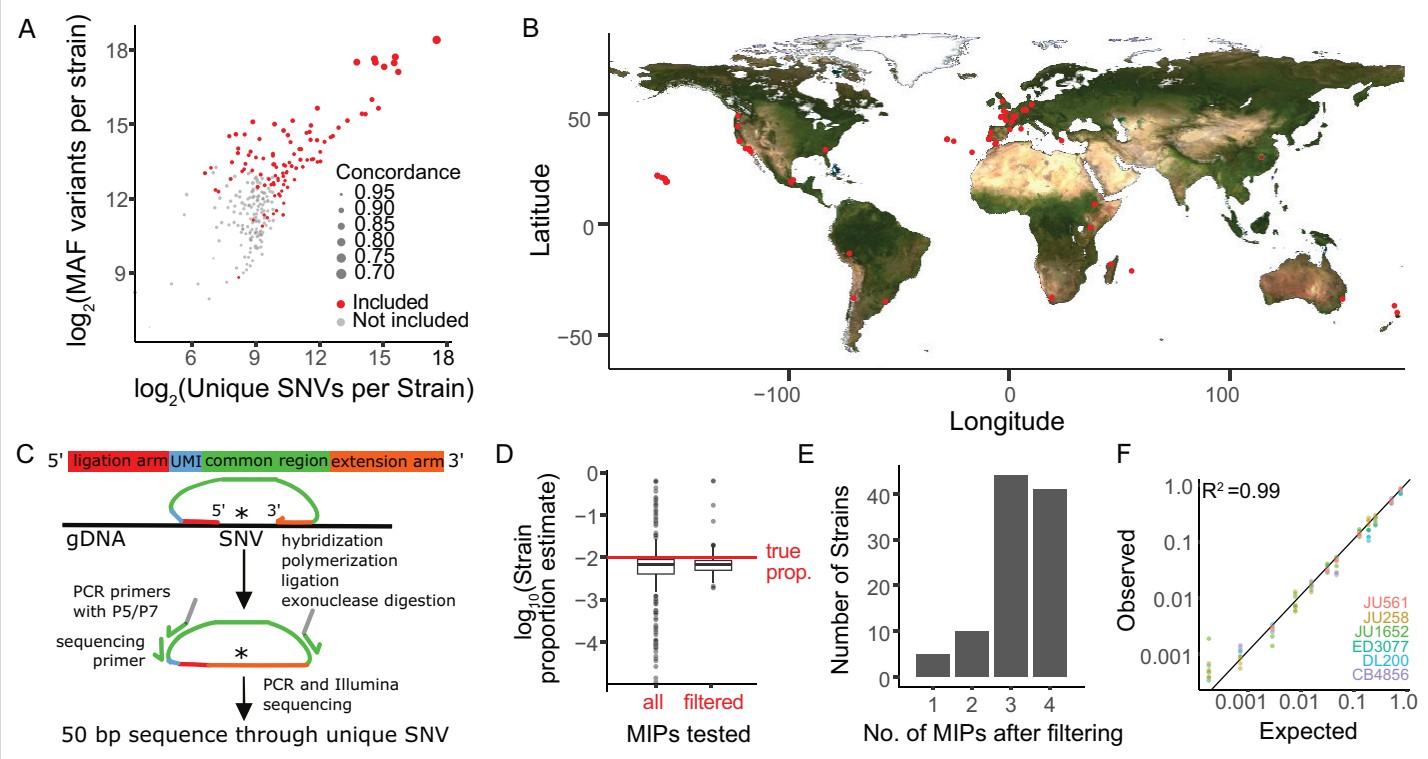

**Figure 1.** Sensitive and precise measurement of strain frequency in pooled culture using MIP-seq. (**A**) The three metrics used to identify the most diverse *C. elegans* strains are plotted. 'MAF' stands for minor allele frequency. Concordance refers to the average pairwise concordance for the focal strain compared to all other strains, which is calculated as the number of shared variant sites divided by the total number of variants for each pair. The strains included in the MIP-seq experiments are in red. (**B**) Geographic locations of the strains assayed for starvation resistance. (**C**) Schematic of MIP-seq. MIPs are designed for loci with SNVs unique to each strain. Four MIPs were designed per strain. MIPs are 80 nt long and include ligation and extension arms to match DNA sequence surrounding the SNV, a unique molecular identifier (UMI), and P5 and P7 sequences for Illumina sequencing. MIPs are hybridized to genomic DNA, polymerized, ligated, and used as PCR template to generate an Illumina sequencing library. The alternative-to-total read frequency for each MIP/SNV locus indicates strain frequency. (**D**) Empirical testing of 412 MIPs with an equimolar mix of genomic DNA from 103 strains to identify reliable MIPs. 321 MIPs passed filtering and were analyzed in the starvation experiment. Outliers for filtered MIPs are for N2, which has hardly any unique SNVs because it is derived from the reference genome. N2 MIPs were included despite poor performance. (**E**) Number of MIPs per strain of the 321 filtered MIPs that passed filtering. (**F**) Genomic DNA from seven strains was combined at different known concentrations, and MIP-seq was used to generate a standard curve. Included MIPs all passed filtering. ($R^2$=0.99).

in the starvation resistance experiment. Three strains with no MIPs passing filtering were excluded from subsequent analysis. As an additional pilot, we mixed genomic DNA from a subset of strains at different concentrations to prepare a standard curve. MIP-seq accurately measured individual strain frequencies over three orders of magnitude (*Figure 1F*), and greater sequencing depth could theoretically expand the dynamic range.

## Using MIP-seq to characterize natural variation of starvation resistance

We used MIP-seq to phenotype 100 diverse strains for starvation resistance. We cultured the strains in standard laboratory conditions, pooled them, and subjected them to starvation during L1 arrest. We aimed for approximately 5000 L1 larvae per strain in the pooled starvation culture in order to ensure representative sampling. However, we expected actual representation to vary across strains and replicates, so we collected DNA from an aliquot of L1 larvae on the first day of starvation as a 'baseline' sample to capture initial population composition. In addition, aliquots were taken from the pool at days 1, 9, 13, and 17 of starvation, and sampled larvae were allowed to recover with food for 4 or 5 days (depending on the duration of starvation), enabling reproduction for 1 day, and then the entire population was collected for DNA preparation (*Figure 2A*). DNA from baseline samples, as well as samples allowed to recover and reproduce following starvation, were sequenced with MIP-seq for each of five biological replicates. It is critical to point out that by incorporating recovery and early

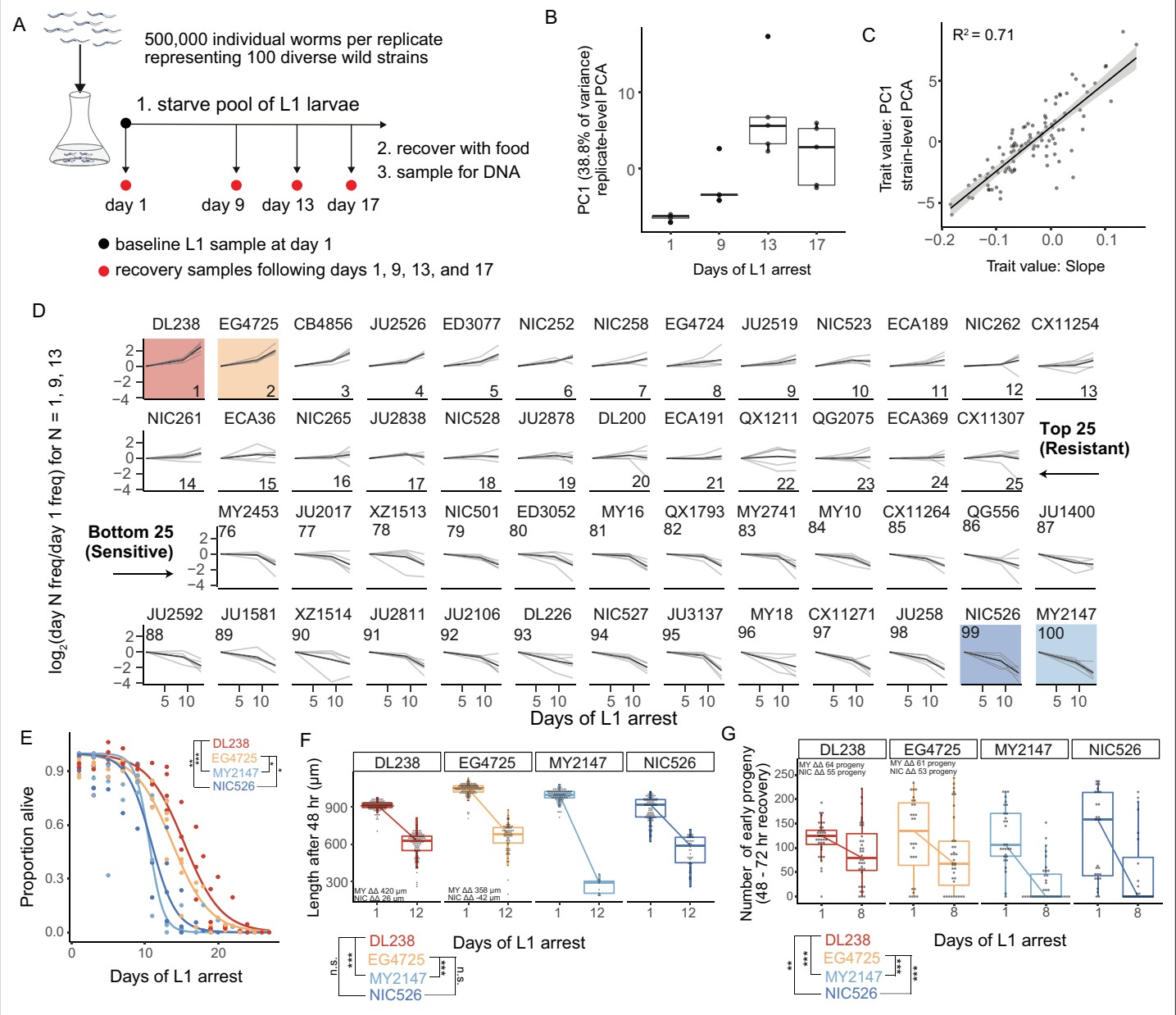

**Figure 2.** MIP-seq determines relative starvation resistance of 100 strains. (**A**) Experimental design. Worms were starved at the L1 stage ('L1 arrest').~5000 L1 larvae per strain were starved (~500,000 total). The population of starved L1 larvae was sampled initially ('baseline' on day 1), and then sampled on the days indicated. Samples (except baseline) were recovered with food in liquid culture, reaching adulthood and producing progeny for 1 day, and the entire population was frozen for DNA isolation. Five biological replicates were performed. (**B**) Principal component 1 of normalized and processed data from all replicates (replicate-level) and strains is plotted, revealing association with duration of starvation. Each point is an individual sample (MIP-seq library). (**C**) The relationship between two starvation-resistance metrics (Slope and PC1) produced from strain-level data (replicates averaged) is plotted. Each point is a different strain. (**D**) Log₂-normalized strain frequency is plotted over time for the 25 most resistant and 25 most sensitive strains in rank order (based on Slope). Only days 1, 9, and 13 are plotted. See *Figure 2—figure supplement 2* for full data. Grey lines are biological replicates and black line is the mean. DL238 and EG4725 are most starvation-resistant, and NIC526 and MY2147 are most sensitive, and they are color-coded accordingly. (**E**) L1 starvation survival curves are plotted for starvation-resistant and sensitive strains. Individual replicate measurements are included as points to which curves were fit with logistic regression. T-tests on 50% survival time of four biological replicates. (**F**) Worm length following 48 hours of recovery with food after 1 or 12 days of L1 starvation. (**G**) Number of progeny produced between 48 and 72 hr of recovery on food following 1 or 8 days of starvation. (**F,G**) ΔΔ indicates effect size of interaction between duration of starvation and strain data plotted in that panel compared to the strain listed (the difference in differences between strains' mean length at days 1 and 12 or between mean number of progeny at days 1 and 8). 'MY' is an abbreviation for MY2147 and 'NIC' is an abbreviation for NIC526. Linear mixed-effects model; one-way p-value of interaction between duration of starvation and strain. (**E–G**) \*\*\*p<0.001, \*\*p<0.01, \*p<0.05.

*Figure 2 continued on next page*

*Figure 2 continued*

The online version of this article includes the following source data and figure supplement(s) for figure 2:

**Source data 1.** Source data for manual starvation resistance assays of wild strains.

**Figure supplement 1.** MIP-seq data analysis and comparison to RAD-seq analysis of starvation resistance.

**Figure supplement 2.** MIP-seq strain frequencies throughout starvation for all strains assayed.

**Figure supplement 3.** Starvation resistance of wild strains is associated with latitude at collection site, but not elevation or substrate.

**Figure supplement 4.** Starvation resistance is negatively correlated with growth rate.

fecundity, this sampling scheme integrates effects of starvation on mortality as well as growth rate and reproductive success, each of which are important for starvation resistance (i.e. fitness) and can be uncoupled by certain genotypes and conditions (*Baugh and Hu, 2020*).

DNA from baseline samples allowed us to effectively normalize differences in pool composition in each replicate, revealing effects of starvation on strain frequency. Differences in pool composition explained the first component in principal component analysis (PCA) when strain frequencies over time were analyzed without consideration of baseline frequencies (*Figure 2—figure supplement 1A*). However, once the data were normalized for initial strain composition using the baseline sample for each replicate, the first principal component correlated with duration of starvation, especially across the first three time points (*Figure 2B*, *Figure 2—figure supplement 1B*). Substantial mortality occurred by day 17 (*Figure 2—figure supplement 1C*), and day 17 recovery cultures thus produced relatively few progeny. Consequently, differences in strain frequencies were actually smaller at day 17 than 13, but relative differences were conserved (*Figure 2—figure supplement 2*). After normalization, duration of starvation is the major factor accounting for differences in strain frequency across all samples, and this is robust to differences in the initial composition of the pool across replicates.

We developed two metrics to quantify relative starvation resistance for each strain. 'Slope' is a measure of how much a strain increases or decreases in frequency over time across days 1, 9, and 13, calculated as the slope of a linear model (*Supplementary file 1*). 'PC1' is the value of the first principal component for each strain from strain-level PCA (*Figure 2B*, *Figure 2—figure supplement 1B*). These two metrics are correlated but also show some differences (*Figure 2C*), suggesting they capture related but also distinct features of the data. While Slope is intuitive, it is limited by the use of a linear model. Nonetheless, Slope values are correlated with starvation-resistance values produced from a previously published population-sequencing approach with less power that included some of the same strains (*Figure 2—figure supplement 1D*; *Webster et al., 2019*). In addition, Slope is modestly correlated with the latitude from which strains were collected, suggesting possible adaptation to starvation or other correlated traits based on location (*Figure 2—figure supplement 3*). There is also a modest negative correlation between Slope and growth rate after only one day of starvation (control condition) (*Figure 2—figure supplement 4*), suggesting a possible trade-off between starvation resistance and population growth rate in the absence of stress. We used the Slope metric to order strains from most resistant to sensitive, revealing differences in starvation resistance between wild strains (*Figure 2D*, *Figure 2—figure supplement 2*). In contrast to Slope, PC1 does not assume linearity, it includes the results from day 17 of starvation, and it may be less affected by noise. PCA is also an established way to obtain trait values for GWA studies (*Ried et al., 2016*; *Yano et al., 2019*).

Our recovery-based sampling approach integrates starvation survival, recovery, and early fecundity into a single fitness assay. It is therefore unclear whether a given strain is more or less resistant because of differences in mortality, growth rate, progeny production, or some combination. It is also unclear what the absolute effect sizes are between the most resistant and sensitive strains in this competition assay. Nonetheless, our approach is intended to model the impact of larval starvation on fitness broadly, while traditional assays can be used to isolate specific effects of starvation on survival, growth, and reproduction in follow-up experiments.

We performed manual assays for starvation survival, growth rate, and early fecundity for the most resistant and sensitive strains. We found starvation-resistant strains DL238 and EG4725 survived starvation significantly longer during L1 arrest than sensitive strains MY2147 and NIC526 (*Figure 2E*). Differences in starvation survival among wild strains are relatively small compared to some published mutants in the N2 reference background (*Baugh and Hu, 2020*). After extended L1 arrest, DL238

and EG4725 recovered from starvation better than MY2147 but not NIC526, as assessed by their size following 48 hr of recovery (*Figure 2F*). Finally, DL238 and EG4725 exhibited a larger early brood size following extended starvation compared to both MY2147 and NIC526. Overall, this demonstrates that differences in starvation resistance among wild strains are driven by differences in survival, recovery, and early fecundity, but that sensitivity of NIC526 is apparently driven by differences in survival and early fecundity without an appreciable effect on growth. These results validate the MIP-seq approach and reveal the extent of natural variation in starvation resistance.

## Natural variation in *irld* gene family members affects starvation resistance

We used Slope and PC1 as trait values to perform GWA using the *Caenorhabditis elegans* Natural Diversity Resource (CeNDR) (*Cook et al., 2017*). GWA identified QTL on the right arm of chromosome IV and on the left and right arms of chromosome V (*Figure 3A–B*, *Supplementary file 2*). We confirmed that each QTL affected starvation resistance by generating NILs and measuring growth rate upon recovery from starvation (*Figure 3—figure supplement 1*). We chose this assay, as opposed to starvation survival or fecundity, because it revealed relatively robust differences between DL238/EG4725 (resistant) and MY2147 (sensitive) (*Figure 2F*).

These QTL are relatively large, ranging from 0.7 to 2.2 Mb, and include many candidate variants (*Supplementary file 2*) across 867 genes, which are enriched for several large gene families. WormCat analysis identified significant enrichments of serpentine receptors, nuclear hormone receptors, and C-type lectins (*Figure 3C*; *Holdorf et al., 2020*). Likewise, protein-domain enrichment analysis (*Finn et al., 2011*) identified seven-pass transmembrane domains and hormone receptor domains (*Figure 3D*). In addition, the receptor L domain was significantly enriched, which is found in proteins comprising Insulin/EGF-Receptor L Domain (IRLD) family (*Dlakić, 2002*). Given weak homology to DAF-2/InsR, and the critical role of IIS in regulation of starvation resistance, we were intrigued at the possibility that natural variation in *irld* family genes may impact starvation resistance, lthough it should be noted that the QTL contain numerous additional candidates that could affect the trait. Across all three QTL identified, there are genetic variants in 16 *irld* genes, and 68 genes have been identified as part of this family in *C. elegans* (*Hobert, 2013*). Multiple variants are present for most *irld* genes, and variants differed in the degree to which they were associated with variation in starvation resistance (*Figure 3E*).

We selected at least one *irld* gene from each QTL for functional analysis. On the left arm of chromosome IV, a variant in *irld-39* was the strongest individual candidate among all genes because of its strong association with starvation resistance and because the variant is predicted to disrupt the start codon of the gene (*Figure 3E and F*, *Supplementary file 2*), likely rendering *irld-39* a functional null in the starvation-resistant strain DL238. However, this was not functionally validated, and it is possible that this variant affects expression of the neighboring *irld* gene, *hpa-1*. On the right arm of chromosome V, *irld-52* was identified through both Slope and PC1 phenotype metrics and contains a variant associated with starvation resistance predicted to disrupt its fifth exon with a frameshift (*Figure 3E and F*), though this was not functionally validated and it is unclear if the variant causes a null mutation. While analyzing variants on the left arm of chromosome V, we noticed that many *irld* genes are adjacent to each other and that each contain many variants. In particular, *irld-11*, *irld-44*, and *irld-45* are clustered, and each gene contains over 50 genetic variants. This pattern of some loci containing many variants relative to N2 has been broadly observed, leading to identification of 'hyper-divergent' regions of the genome containing exceptional amounts of variation (*Lee et al., 2021*). *irld-11*, *irld-44*, and *irld-45* are part of a hyper-divergent region, and because they are so tightly linked, they are hyper-divergent in the same strains. We found that hyper-divergence at these loci was associated with starvation resistance (*Figure 3G*). *irld-57* is also in a hyper-divergent region on the right arm of chromosome V, and hyper-divergence at this locus is also associated with starvation resistance (*Figure 3G*). Given several variants predicted to disrupt protein function in each, we believe *irld-11* and *irld-57* are null in the hyper-divergent context, though this has not been functionally demonstrated. Notably, associations between variants or hyper-divergence and Slope (*Figure 3F and G*) together with their predicted negative impacts on gene function (*Figure 3H*) suggests that disruption of these four *irld* genes in backgrounds where they are functional will increase starvation resistance.

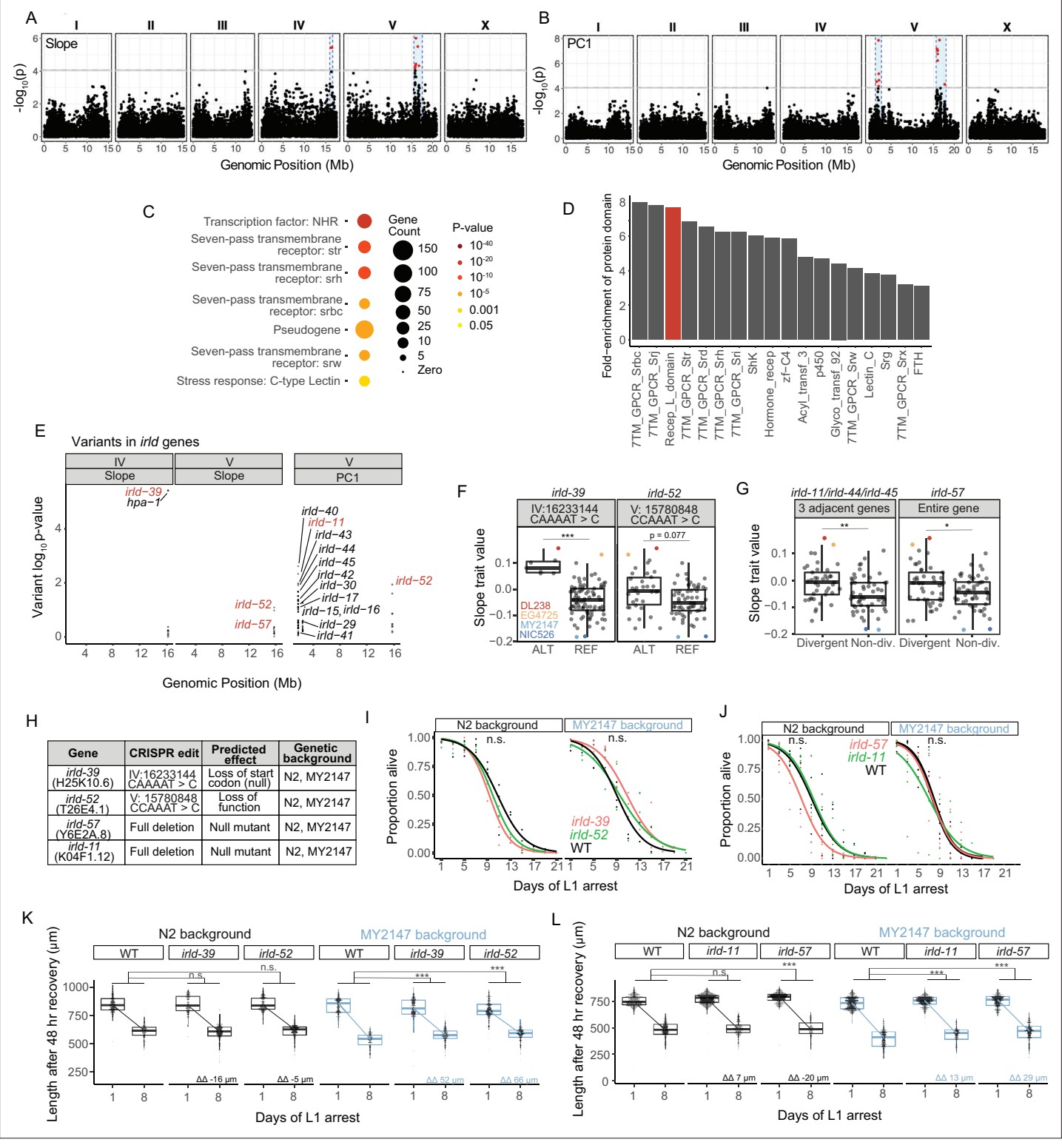

**Figure 3.** Genetic variation in the *irld* gene family underlies differences in starvation resistance. (**A**) GWA output using Slope as a trait value. Significant QTL intervals are IV: 15939340–16613710 and V: 15660911–17615557. (**B**) GWA output using PC1 as a trait value. Significant QTL intervals are V: 1345848–2764788 and V: 15775895–18065050. (**C**) WormCat Category 3 enrichments for all genes with variants in the QTL. (**D**) Fold-enrichment of protein domains significantly enriched among genes with variants in QTL. A hypergeometric p-value was calculated for each of 102 protein domains present, and a Bonferroni-corrected p-value of 0.00049 was used as a threshold to determine significance. Red indicates the receptor L domain, which is found in *irld* genes. (**E**) All variants in *irld* genes that are within significant QTL and their association with the starvation-resistance traits, Slope and PC. Each gene

*Figure 3 continued on next page*

*Figure 3 continued*

name is shown next to the most significant variant for that gene, but multiple variants are plotted for each gene when present. Red indicates genes selected for functional validation. (**F**) Slope trait values for strains based on whether they have ALT and REF alleles for specific *irld-39* and *irld-52* variants predicated to disrupt protein function. The *irld-52* variant p-value is p=0.007 for the PC1 trait value (only the Slope trait value is shown here). Significance determined from GWA fine mapping. (**G**) Slope trait values for strains based on whether they are hyper-divergent or not at *irld-11* and *irld-57* loci. T-test on trait values between hyper-divergent and non-divergent strains. (**F–G**) DL238, EG4725, NIC526, and MY2147 are color-coded as indicated. (**H**) The four *irld* genes selected for genome editing and the edits generated for each. For *irld-39 and irld-52,* N2 and MY2147 have the REF allele and were edited to have the ALT allele. *irld-11* and *irld-57* are hyper-divergent in DL238 and EG2745 backgrounds, so full gene deletions were generated in N2 and MY2147 backgrounds. (**I**) L1 starvation survival assays on *irld-39* and *irld-52* ALT alleles in N2 and MY2147 backgrounds. There were no significant differences between strains within a background. (**J**) L1 starvation survival assays on *irld-11* and *irld-57* deletions in N2 and MY2147 backgrounds. There were no significant differences between strains within a background. (**K–L**). Worm length following 48 hr recovery with food after 1 or 8 d of L1 starvation for indicated genotypes. Linear mixed-effects model; one-way p-value for interaction between strain and duration of starvation; 4–5 biological replicates per condition. ΔΔ indicates effect size of interaction between duration of starvation and strain compared to control (the difference in differences between strains' mean length at days 1 and 8). (**F,G,K,L**) ***p<0.001, **p<0.01, *p<0.05, n.s. not significant.

The online version of this article includes the following source data and figure supplement(s) for figure 3:

**Source data 1.** Source data for starvation resistance assays of *irld* strains.

**Figure supplement 1.** Validation of QTL with sequenced near-isogenic lines (NILs).

**Figure supplement 1—source data 1.** Source data for worm length measurements of generated NILs.

We used CRISPR-Cas9 genome editing to determine functional consequences of genetic modification of our candidate *irld* genes. Because *irld-39* and *irld-52* contain singular variants associated with starvation resistance and predicted to disrupt protein function, we generated these specific variants in the starvation-sensitive MY2147 and the laboratory-reference N2 backgrounds (*Figure 3H*). Since *irld-11* and *irld-57* contain so many candidate variants, we deleted these genes in MY2147 and N2, rendering them null at each locus (*Figure 3H*). Edits of *irld-39* and *irld-52* are more likely to approximate the effect of specific variants in the wild, because they are the exact variants present in starvation-resistant wild strains. None of the alleles in either background significantly affected survival (*Figure 3I and J*). A power analysis suggests there is sufficient statistical power to detect differences of approximately 2 days or greater, suggesting there is not a difference of at least this magnitude. However, alleles for all four *irld* genes mitigated the effect of starvation on growth rate in the MY2147 background but not N2 (*Figure 3K and L*). This suggests that MY2147, as a more starvation-sensitive background than N2, facilitates detection of alleles that increase starvation resistance. These results show that multiple types of variants in different *irld* family members reduce the effect of extended L1 starvation on recovery, suggesting four individual genes from this family affect this aspect of starvation resistance in wild strains. Notably, none of the engineered variants affected the trait to a similar extent as the NILs, suggesting that other variants within each QTL also affect the trait.

## IRLD-39 and IRLD-52 act through DAF-16/FoxO

We hypothesized that *irld-39* and *irld-52* have additive phenotypic effects, and that combining our two engineered alleles would reveal an effect in N2. An *irld-39(duk1); irld-52(duk17)* double mutant did not significantly increase starvation survival in the N2 background (*Figure 4A*). In this case, there was sufficient statistical power to detect differences of approximately 1.5 days or greater. However, the double mutant displayed a modest but significant increase in growth following 8 days of starvation, consistent with single mutants in the MY2147 background (*Figure 4B*). Furthermore, the double mutant significantly increased early fecundity following starvation (*Figure 4C*). These results further support the conclusion that natural variation in *irld-39* and *irld-52* affects starvation resistance. Notably, these two variants are both present in the most starvation-resistant strain identified, DL238 (*Figure 3F*).

Given weak homology between IRLD proteins and the extracellular domain of DAF-2/InsR, we wondered if IRLD-39 and IRLD-52 modify IIS, as originally proposed (*Dlakić, 2002*). We therefore hypothesized that increased starvation resistance with disruption of *irld-39* and *irld-52* depends on *daf-16/FoxO*. Again, the *irld-39; irld-52* double mutant displayed significant mitigation of the effect of starvation on growth (*Figure 4D*). This result corroborates the effect of the double mutant after

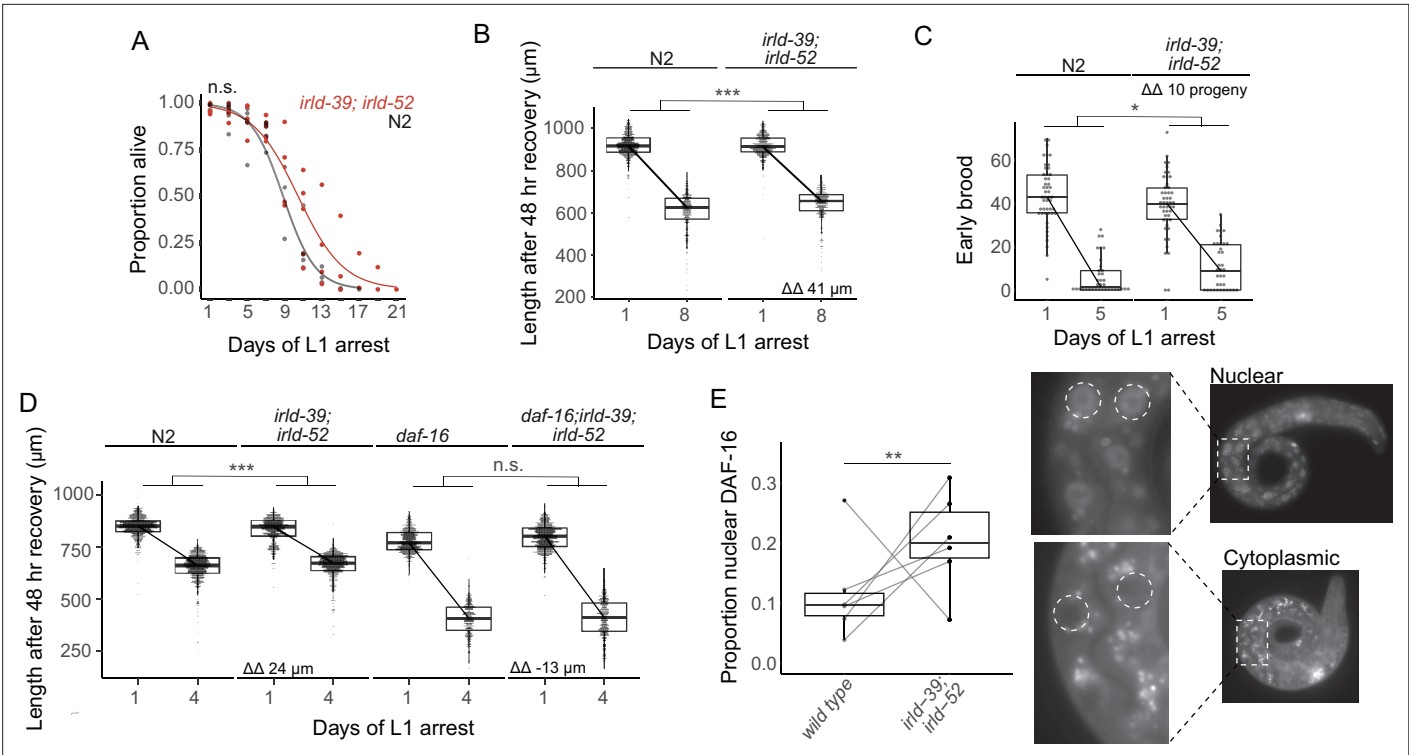

**Figure 4.** IRLD-39 and IRLD-52 together impact starvation resistance and depend on DAF-16. (**A**) Survival curves of *irld-39(duk1); irld-52(duk17)* and N2 throughout L1 starvation. The apparent increase in starvation survival in the double mutant is not statistically significant (*P*=0.14). (**B**) Worm length of *irld-39(duk1); irld-52(duk17)* and N2 following 48 hr of recovery with food after 1 or 8 days of L1 starvation. (**C**) Number of progeny produced between 48 and 72 hr of recovery with food after 1 or 5 days of L1 starvation. (**D**) Worm length of N2, *irld-39(duk1); irld-52(duk17)*, *daf-16(mu86)*, and *daf-16(mu86); irld-39(duk1); irld-52(duk17)* following 48 hr of recovery with food after 1 or 4 days of L1 starvation. (**B–D**) Linear mixed-effects model with duration of L1 starvation and genotype as fixed effects and the number of replicates as a random effect; p-value calculated for interaction between fixed effects. ΔΔ indicates effect size of interaction between duration of starvation and strain compared to control. (**E**) Nuclear localization of DAF-16::GFP in intestinal cells of starved L1s ~36 hr after hatching. Each point represents the result of a single independent biological replicate with 51–64 worms scored for each condition and replicate, with a line connecting the two genotypes in each replicate. The Cochran-Mantel-Haenszel test was used to determine differences in the distribution of the two categories (nuclear and cytoplasmic) between *daf-16(ot971)* (wild type) and *daf-16(ot971); irld-39(duk1); irld-52(duk17)* (*irld-39; irld-52*). Images of intestinal nuclear and cytoplasmic localization are shown. (**A–E**) Four to six biological replicates were performed per experiment. \*\*\*p<0.001, \*\*p<0.01, \*p<0.05, n.s. not significant.

The online version of this article includes the following source data and figure supplement(s) for figure 4:

**Source data 1.** Source data for starvation resistance assays of *irld-39(duk1); irld-52(duk17)*.

**Figure supplement 1.** *irld* genes are up-regulated in starved L1s compared to fed L1s.

**Figure supplement 2.** *irld* genes exhibit a bias toward expression in sensory neurons with some expression in other cell types.

8 days of starvation (*Figure 4B*), except after only 4 days in this case (4 days of starvation was used since the *daf-16* mutant is starvation-sensitive). We found no significant difference in the effect of starvation on growth between the null mutant *daf-16(mu86)* and *daf-16(mu86); irld-39(duk1); irld-52(duk17)*, suggesting that increased starvation resistance of *irld-39(duk1); irld-52(duk17)* is dependent on *daf-16* (*Figure 4D*). This genetic epistasis is consistent with DAF-16/FoxO activity being increased in the *irld-39(duk1); irld-52(duk17)* double mutant. In support of this hypothesis, nuclear localization of endogenous DAF-16 (*Aghayeva et al., 2020*) in intestinal cells, a proxy of its activity, was significantly increased in *irld-39(duk1); irld-52(duk17)* mutants (*Figure 4E*). However, this is a relatively modest difference in nuclear localization, and it is unclear where in the animal DAF-16 activity is most relevant in this context. Nonetheless, genetic epistasis and nuclear localization assays suggest that IRLD-39 and IRLD-52 act through DAF-16/FoxO to affect starvation resistance during L1 arrest.

## Discussion

Our results illustrate the power of MIP-seq as a population selection-and-sequencing approach for analysis of complex traits in *C. elegans*. MIP-seq can be used in any organism with known sequence variants and that can be cultured in sufficiently large numbers with the ability to select on the trait of interest. With sufficient population genetic complexity and sequencing depth, meaningful phenotypic differences too small or variable to be detected by manual assays can be discovered, leading to improved understanding of gene-by-environment interactions and the genotype-to-phenotype map. When complex traits are highly polygenic (*Boyle et al., 2017*), it is critical to leverage the power of sequencing to elucidate their architectures. Here we used MIP-seq with a large panel of wild strains for statistical genetic analysis, but it can also be used with panels of recombinant lines for high-resolution gene mapping (*Mok et al., 2017*). MIP-seq can also be used for phenotypic analysis of mutants where it is beneficial to boost sensitivity and precision by using sequencing to count exceptionally large numbers of individuals (*Shendure et al., 2017*; *Mok et al., 2020*).

We characterized natural variation in starvation resistance in a set of genetically diverse, wild strains of *C. elegans* using MIP-seq and traditional assays. Our results suggest relatively little phenotypic variation of this presumably fitness-proximal trait. Nonetheless, we validated three QTL and showed four *irld* genes in these QTL impact starvation recovery. For *irld-11* and *irld-57*, we generated deletion mutants, which do not precisely match the variants present in wild strains. For *irld-39* and *irld-52*, the engineered alleles match starvation-resistant strains, but we have not confirmed their loss of function. Thus, our results suggest, but do not definitively demonstrate, that variation in *irld* genes affects starvation resistance in this species. The *irld* gene family is expanded relative to other *Caenorhabditis* species, suggesting that expansion (or contraction) of gene families influences natural variation and possibly evolutionary adaptation in this context. In addition, two of the *irld* genes identified are in hyper-divergent regions of the genome, consistent with genes in these regions contributing to environmental responses (*Lee et al., 2021*). However, *irld* variants investigated each had relatively weak phenotypic effects compared to the NILs, suggesting they do not fully account for natural variation in the trait associated with the QTLs. This implies other variants (*Supplementary file 2*), possibly of larger effect, also contribute to phenotypic variation.

Genetic epistasis analysis suggests that the effect of *irld-39* and *irld-52* on starvation resistance depends on *daf-16/FoxO*, and the double mutant increases DAF-16 nuclear localization, suggesting that these *irld* genes modify IIS. However, *irld-39* and *irld-52* could affect DAF-16 activity independent of IIS and could also affect other signaling pathways. IRLDs also bear weak homology to EGF receptors, and *irld* family members *hpa-1* and *hpa-2* affect healthspan by modifying EGF signaling (*Iwasa et al., 2010*). It is not known whether EGF signaling affects starvation resistance or other aspects of L1 arrest, and future work is needed to address the possible role of *irld* genes affecting EGF signaling in this context.

### Ideas and speculation

Given the proposal that IRLD proteins modify IIS, it is intriguing to speculate that they do so by binding any of the 40 insulin-like peptides (ILPs) that would otherwise agonize or antagonize DAF-2/InsR (*Pierce et al., 2001*), as suggested previously (*Dlakić, 2002*). DAF-2B is an alternative isoform of DAF-2/InsR that includes the extracellular domain but lacks the tyrosine kinase domain, like the IRLD proteins, and it is also thought to act this way (*Martinez et al., 2020*). This hypothetical mechanism is also analogous to the proposed function of insulin-like growth factor (IGF)-binding proteins, which affect circulation and receptor binding of IGF proteins (*Allard and Duan, 2018*). These parallels suggest the possibility that natural variation in the IGF-binding protein family (*Rotwein, 2017*) contributes to phenotypic variation in humans. However, we have not shown that IRLD proteins actually bind ILPs, and a variety of uncertainties remain regarding their function.

Expression analysis provides clues to how *irld* genes possibly influence starvation resistance. Published whole-animal mRNA-seq analysis of fed and starved L1 larvae (*Webster et al., 2018*) revealed relatively low expression levels of the entire *irld* family (*Figure 4—figure supplement 1*). However, about half of the *irld* genes were differentially expressed, and all of those were upregulated in starved larvae, suggesting a role in starvation. We also interrogated existing single-cell RNA-seq datasets. One includes the major tissue types in fed L2-stage larvae (*Cao et al., 2017*), and it suggests that *irld* genes are most prominently expressed in ciliated sensory neurons, though there is expression

in other neurons and tissues (*Figure 4—figure supplement 2*). Another study focused on neurons in fed L4-stage larvae (*Taylor et al., 2021*), and it suggests that *irld* gene expression is more prominent in sensory neurons than other neuron types (*Figure 4—figure supplement 2*). *irld-39* is expressed in ASJ sensory neurons, along with distal tip cells and vulval precursors (*Figure 4—figure supplement 2*). *irld-52* is expressed in the ADL sensory neurons and also intestinal rectal muscle cells. *C. elegans* sensory neurons are polymodal and influence life-history traits regulated by IIS, including dauer formation, aging, and L1 arrest (*Bargmann and Horvitz, 1991*; *Vowels and Thomas, 1992*; *Apfeld and Kenyon, 1999*). ASJ is known to express the relatively potent ILP DAF-28 in nutrient and sensory-dependent fashion (*Li et al., 2003*; *Kaplan et al., 2018*), and *daf-28* affects L1 starvation survival (*Chen and Baugh, 2014*). *ins-4/ILP* is also expressed in ASJ, and it too affects L1 starvation survival (*Chen and Baugh, 2014*). If IRLD-39 and IRLD-52 proteins are translated and function in the vicinity of these sensory neurons, that would allow them to exert their influence at the interface of the animal and its environment.

## Materials and methods
### Strains used in this study

In addition to N2, wild isolates CB4854, CB4856, CX11254, CX11264, CX11271, CX11276, CX11285, CX11307, DL200, DL226, DL238, ED3049, ECA189, ECA191, ECA36, ECA363, ECA369, ECA372, ECA396, ED3017, ED3052, ED3077, EG4724, EG4725, GXW1, JU1212, JU1400, JU1581, JU1652, JU1793, JU1896, JU2001, JU2007, JU2017, JU2106, JU2234, JU2316, JU2464, JU2519, JU2526, JU2576, JU258, JU2592, JU2619, JU2811, JU2829, JU2593, JU2838, JU2841, JU2878, JU2879, JU3137, JU561, JU774, JU775, JU782, KR314, LKC34, MY10, MY16, MY18, MY2147, MY23, MY2453, MY2741, NIC195, NIC199, NIC251, NIC252, NIC256, NIC527, NIC258, NIC261, NIC262, NIC265, NIC266, NIC268, NIC271, NIC3, NIC501, NIC523, NIC526, NIC528, PB306, PS2025, QG2075, QG556, QW947, QX1211, QX1212, QX1791, QX1792, QX1793, QX1794, WN2001, XZ1513, XZ1514, XZ1515, and XZ1516 were phenotyped for starvation resistance using MIP-seq. In addition to these 100 strains, CX11262, ECA348, and NIC260 were included in the MIP-seq pilot but excluded from subsequent analysis based on quality-control metrics described in the 'MIP-seq analysis' section. QX1430 was used for validation assays. All wild isolates were obtained from CeNDR (*Cook et al., 2017*). CB1370 *daf-2(e1370) III,* CF1038 *daf-16(mu86) I,* and OH16024 *daf-16(ot971[daf-16::GFP]) I* were used to assess the interaction of *irld* genes with insulin signaling.

### Strains generated in this study

Near-isogenic lines include:

> LRB392 – *dukIR7(V, EG4725 >MY2147)*
> LRB393 – *dukIR8(V, EG4725 >MY2147)*
> LRB395 – *dukIR10(V, EG4725 >MY2147)*
> LRB396 – *dukIR11(V, MY2147 >EG4725)*
> LRB397 – *dukIR12(V, MY2147 >EG4725)*
> LRB398 – *dukIR13(V, MY2147 >EG4725)*
> LRB399 – *dukIR14(V, MY2147 >EG4725)*
> LRB400 – *dukIR15(V, MY2147 >EG4725)*
> LRB401 – *dukIR16(V, MY2147 >EG4725)*
> LRB402 – *dukIR17(V, EG4725 >MY2147)*
> LRB403 – *dukIR18(V, EG4725 >MY2147)*
> LRB407 – *dukIR19(V, EG4725 >MY2147)*
> LRB408 – *dukIR20(V, EG4725 >MY2147)*
> LRB409 – *dukIR21(V, EG4725 >MY2147)*
> LRB410 – *dukIR22(IV, DL238>N2)*
> LRB411 – *dukIR23(IV, DL238>N2)*

See *Figure 3—figure supplement 1* for wild isolate composition.
CRISPR-edited strains and new crosses include:

> LRB412 *irld-39* in N2 background – *irld-39(duk1) IV*
> LRB413: *irld-39* in MY2147 background – *irld-39(duk2*[MY2147]*) IV*

LRB414: *irld-39* in MY2147 background – *irld-39(duk3*[MY2147]) *IV*
LRB415: *irld-39* in MY2147 background – *irld-39(duk4*[MY2147]) *IV*
LRB420: *irld-11* in MY2147 background – *irld-11(duk9*[MY2147]) *V*
LRB421: *irld-52* in N2 background – *irld-52(duk10) V*
LRB422: *irld-52* in MY2147 background - *irld-52(duk11*[MY2147]) *V*
LRB423: *irld-11* in N2 background – *irld-11(duk12) V*
LRB425: *irld-57* in N2 background – *irld-57(duk13) V*
LRB426: *irld-57* in N2 background – *irld-57(duk14) V*
LRB427: *irld-57* in MY2147 background – *irld-57(duk15*[MY2147]) *V*
LRB428: *irld-57* in MY2147 background – *irld-57(duk16*[MY2147]) *V*
LRB431: *irld-52* in N2 background – *irld-52(duk17) V*
LRB444: *irld-39; irld-52* (generated from crossing LRB412 and LRB431) - *irld-39(duk1) IV; irld-52(duk17) V*
LRB456: *daf-16(mu86) I; irld-39(duk1) IV; irld-52(duk17) V*
LRB457, LRB458: *daf-2(e1370) III; irld-39(duk1 IV); irld-52(duk17) V*
LRB463: *daf-16(ot971) I; irld-39(duk1) IV; irld-52(duk17) V*

Multiple strain names for the same genotype indicates independent lines.

## MIP-seq experimental set-up

Wild strains were independently passaged on 10 cm NGM plates with OP50 *E. coli* every two to three days to ensure they did not starve for at least three generations prior to the experiment. For each biological replicate, a single non-starved plate with gravid adults was selected per strain to ensure initial representation of all strains. Strains were pooled for hypochlorite treatment to obtain pure populations of embryos (*Hibshman et al., 2021*). Embryo concentration was calculated by repeated sampling, and 500,000 embryos were resuspended at 10/µL in S-basal (50 mL total culture) and placed in a 20°C shaker at 180 rpm to hatch without food and enter L1 arrest. On day 1 (24 hr after hypochlorite treatment), 5 mL of culture (50,000 L1s) was taken as a baseline sample, spun down at 3000 rpm, aspirated down to approximately 100 µL in an Eppendorf tube, flash frozen in liquid nitrogen, and stored at –80°C until DNA isolation. At days 1, 9, 13, and 17, aliquots from the L1 arrest culture were set up in recovery cultures at 5 L1s/µL, 1x HB101 (25 mg/mL), and S-complete. Recovery cultures were 10 mL for days 1 and 9, 20 mL for day 13, and 50 mL for day 17 to account for lethality late in starvation by ensuring adequate population sizes. Four days after recovery culture set-up, samples were collected for DNA isolation. For days 1 and 9, the recovery culture was freshly starved with adults and next-generation L1 larvae. At day 13, the culture was typically near starved, with adults and some L1 larvae. At day 17, the culture was typically not starved. If HB101 was still present at collection, samples were washed 3–4 times with S-basal. Samples were flash frozen in liquid nitrogen and stored at –80°C until DNA isolation.

## DNA isolation

Frozen samples were rapidly freeze-thawed three times, cycling between liquid nitrogen and a 45°C water bath. Genomic DNA was isolated using the Quick-DNA Miniprep Kit (Zymo Research# D3024) following the manufacturer's protocol. The DNA concentration was determined for each sample using the Qubit dsDNA HS Assay kit (Invitrogen# Q32854).

## MIP design

MIPgen (*Boyle et al., 2014*) was used to design four MIPs for each of 103 strains. Unique homozygous SNVs were parsed from the VCF file WI.20170531.vcf.gz (available at https://storage.googleapis.com/elegansvariation.org/releases/20170531/WI.20170531.vcf.gz). Target regions in BED format were generated using the makeBedForMipgen.pl script. MIPgen was used against *C. elegans* genome version WS245 with the following parameters: -min_capture_size 100 -max_capture_size 100 -tag_sizes 0, 10. MIPs are 80 base-pairs (bp) long and include 20 bp ligation and extension arms that are complementary to DNA surrounding the unique SNV of interest for each strain. In addition, P5 and P7 Illumina sequences are included as part of the MIP to facilitate Illumina sequencing. Each MIP molecule includes a 10 bp unique molecular identifier (UMI) adjacent to the ligation arm. Only MIPs that capture the SNV within a 50 bp sequencing read were used, meaning the SNV was no more than 40 bp away from the UMI. SNPs located within 40 bases of

the sequencing start site were parsed with the parseMipsPerSNPposition.pl script. These scripts can be found at https://github.com/amykwebster/MIPseq_2021 (*Webster, 2021*; copy archived at swh:1:rev:27839dcc9ef1587086be195349310fb70fbfcaf1).

## MIP-seq library preparation and sequencing

For pilots and the starvation-resistance experiment, 500 ng genomic DNA from each sample was used for MIP-seq libraries. Libraries were generated as described previously (*Hiatt et al., 2013*) with the following modifications. We included 1,000 copies of each MIP for every individual copy of the worm genome in the 500 ng input DNA, which corresponded to 0.0083 picomoles of each individual MIP. All 412 MIPs (sequences available in *Supplementary file 1*) were first pooled in an equimolar ratio at a concentration of 100 μM. The MIP pool was diluted to 5 μM in 1 mM Tris buffer, and 50 μL of this pool was used in the 100 μL phosphorylation reaction. Next, the probe hybridization reaction for each sample was set with 500 ng DNA and 3.42 picomoles (0.0083 picomoles x 412) of the phosphorylated probe mixture. Following hybridization, gap filling, ligation, and exonuclease steps were performed as described previously. PCR amplification of the captured DNA (primer sequences available in *Supplementary file 1*) was performed in a 50 μL reaction with 18 cycles. The PCR libraries were purified using the SPRIselect beads (Beckman# B23318), and library concentrations were assessed with the Qubit dsDNA HS Assay kit (Invitrogen# Q32854). Sequencing was performed on the Illumina HiSeq 4000 to obtain 50 bp single-end reads.

## MIP-seq analysis

FASTQ files from sequencing reads were processed using the script parseMIPGenotypeUMI.pl, also available at https://github.com/amykwebster/MIPseq_2021. This script accepts as input the list of MIPs produced from MIPgen, the UMI length, and FASTQ files in order to count the number of reads corresponding to each MIP and whether they have the reference allele, alternative allele, or one of two other alleles. While we included UMIs in our MIP design, use of the UMI to filter duplicate reads in pilot standard curves did not improve data quality (likely due to the relatively large mass of DNA used to prepare libraries), and so the UMI was not used in the published analysis. For each MIP, the frequency of the strain for which the MIP captures its unique SNV was calculated as the alternative read count divided by the total of alternative and reference read counts. For the MIP pilot with all strains in an equimolar ratio, there were 246,986,236 total mapped reads to all MIPs. Individual MIPs were filtered out if they did not meet the following criteria: (1) They were within 3.5-fold of expected frequency (that is, alt / (alt +ref) was within 3.5-fold of 1/103), (2) 'other' reads (those that are not alternative or reference alleles) were <20,000 total, and (3) alternative and reference allele totals were between 20,000 and 2,000,000 total reads. 321 of 412 MIPs met these criteria, and reads from these 321 MIPs were included in subsequent analysis. N2 has very few unique SNVs making it difficult to design optimal MIPs, and N2 MIPs did not meet these criteria but were included nonetheless (see *Supplementary file 1*). For the standard curve experiment (*Figure 1F*), DNA from seven strains (CB4856, DL200, ED3077, JU258, JU561, JU1652, and N2) was pooled in defined concentrations ('expected'), and MIPs that met the criteria defined above were used to calculate strain frequencies ('observed', see *Supplementary file 1*).

For the starvation-resistance experiment, an average of 51.7 million reads (standard deviation 7.2 million reads) were sequenced per library (one library per time point, replicate, and condition – 25 libraries total). An average of 94% of reads (standard deviation 0.5%) matched the ligation probe, and 71.8% (standard deviation 3.4%) matched the ligation probe and scan sequence. Strain frequencies were determined by averaging the frequencies calculated across MIPs included in the 321 MIPs for each strain. A dataframe of all strains and their frequencies at day 1 baseline, as well as days 1, 9, 13, and 17 after recovery for all replicates was used to obtain trait values for subsequent analysis. PCA was performed on the dataframe following normalization of day 1, 9, 13, and 17 time points by the baseline day 1 sample and log2 transformation. PC1 loadings were extracted for each strain. For 'Slope', day 1, 9, and 13 recovery samples were normalized by day 1 frequencies and log2 transformed. For each strain, a line was fit to day 1, 9, and 13 normalized data with intercept at 0, and the slope of the line was taken as the trait value.

## Comparison of MIP-seq and RAD-seq

To determine how well MIP-seq trait values correlated with RAD-seq trait values from previous work (*Webster et al., 2019*), RAD-seq data were normalized the same way that we normalized the MIP-seq data. Specifically, data from one biological replicate from RAD-seq that had data at time points over the course of starvation, including days 1, 7, 14, 21, and 24, was used. The frequency of each strain at each time point was divided by its frequency on day 1. These values were log2 transformed, so positive values indicate an increase in frequency over time and negative values indicate a decrease in frequency over time. A linear regression was then fit to each with a y-intercept of 0 through the data points over time. The slope of the line was calculated as the trait value for each strain. RAD-seq and MIP-seq data were filtered to include only the 34 strains that were present in both analyses. The values were plotted against each other and a linear regression was fit through these points to determine their correlation ($R^2$=0.24, p=0.002).

## GWA analysis

Slope and PC1 trait values for each strain were used for GWA using the R package cegwas2 (*Zdraljevic et al., 2021*). Genotype data were acquired from the latest VCF release (release 20200815) from CeNDR. BCFtools (*Li, 2011*) was used to filter variants below a 5% minor allele frequency and variants with missing genotypes and used PINKv1.9 (*Purcell et al., 2007*; *Chang et al., 2015*) to prune genotypes using linkage disequilibrium. The additive kinship matrix was generated from 45,733 markers using the A.mat function in the rrBLUP package. Because these markers have high LD, eigen decomposition of the correlation matrix of the genotype matrix was performed to identify 570 independent tests. GWA was performed using the GWAS function of the rrBLUP package (*Endelman, 2011*). Significance was determined by an eigenvalue threshold by the number of independent tests in the genotype matrix. Confidence intervals were defined as +/-150 SNVs from the rightmost and leftmost markers passing the significance threshold.

ALT and REF information for *irld-39* and *irld-52* high-impact variants was obtained from fine mapping and is available as part of *Supplementary file 2*. To determine whether *irld-57* and *irld-11* overlapped with hyper-divergent regions in each strain, coordinates of hyper-divergent regions for each strain were obtained from *Lee et al., 2021*, and coordinates of *irld-11* and *irld-57* were obtained from WormBase. If the hyper-divergent region and gene overlapped for a strain, then the strain was considered hyper-divergent at the locus. Hyper-divergent status of each strain is available in *Supplementary file 2*.

## Enrichment analyses

To identify enriched gene groups, fine mapping data from Slope and PC1 results were merged with WS273 gene names. Unique sequence names were extracted (see *Supplementary file 2*), and the 867 sequence names with variants in Slope or PC1 QTL were used in WormCat (*Holdorf et al., 2020*) to identify functional category enrichments (*Figure 3C*). The most specific enrichments (those in Category 3) are shown.

For protein domain enrichment analysis, a protein fasta file was downloaded from Wormbase (c_elegans.PRJNA13758.WS281.protein.fa). To determine enriched protein domains, the 867 sequence names present among genes with variants in significant QTL were first used to subset this fasta file. In cases in which a gene had multiple versions within the fasta file, the 'a' isoform of the gene was used. 644 of the 867 sequence names had protein sequences in the fasta file; most others are annotated as pseudogenes and presumably do not have protein sequences. The protein sequences were used as input using the hmmscan program (https://www.ebi.ac.uk/Tools/hmmer/search/hmmscan) and searching the Pfam database (*Finn et al., 2011*). To obtain a background set of protein domains, the genome fasta file was also used to search the Pfam database. Fasta files were split into groups of 500 sequences that are between 10 and 5,000 peptides to comply with the hmmscan search algorithm. To calculate enrichment of protein domains, hypergeometric p-values were calculated for each protein domain present among genes with variants in significant QTL. 102 protein domains were present, so a Bonferroni-corrected p-value of 0.00049 was used as a significance threshold. Protein domains were excluded if the domain was not present at least five times among genes with variants in significant QTL.

## NIL generation

To validate chromosome IV and V QTL, pairs of strains that differ for starvation resistance and the alternative vs reference allele for the associated SNV marker were chosen. Compatibility at the *peel-1/zeel-1* and *pha-1/sup-35* loci was considered (**Seidel et al., 2008**; **Ben-David et al., 2017**). For chromosome V QTL, EG4725 and MY2147 were compatible at both loci, and we generated reciprocal NILs for the left and right arms of chromosome V. EG4725 did not have the alternative allele associated with starvation resistance for chromosome IV, so we used DL238 and N2 as the parental strains. DL238 and N2 are incompatible for reciprocal crosses, but we introgressed the DL238 chromosome IV QTL into the N2 background. To generate NILs, the two parental strains were first crossed, then F2 progeny were genotyped on each end of the desired QTL for introgression to identify homozygotes from one parental background (e.g. MY2147). Then these homozygotes were repeatedly backcrossed to the opposite background (e.g., EG4725) and repeatedly genotyped to maintain homozygotes at the introgressed region. Genotyping was performed using PCR to amplify a genomic region whose sensitivity to a particular restriction enzyme depends on parental genetic background. Primers were designed using VCF-kit (**Cook and Andersen, 2017**). Primers and enzymes used can be found in **Supplementary file 2**. NILs were backcrossed a minimum of six times. Final NILs were sequenced at ~1 x coverage to determine the parental contributions over the entire genome (**Figure 3—figure supplement 1** and **Supplementary file 2**).

## CRISPR design and implementation to edit *irld* genes

For genes of interest, CRISPR guide design was done in Benchling using genome version WBcel235 and importing sequence for genes of interest. To generate *irld-39* and *irld-52* variants (5 bp deletions), a single guide RNA (sgRNA) and repair template was generated. For *irld-11* and *irld-57,* two sgRNAs were generated per gene to delete the entire gene and a single repair template was used. sgRNAs (2 nmol) and 100 bp repair templates (highest purity at 4 nmol) were ordered from IDT. The *dpy-10* co-CRISPR method was used to generate and screen for edits (**Paix et al., 2015**). The injection mix used was: sgRNA for *dpy-10* (0.2 µL of 100 µM stock), sgRNA of gene of interest (0.5 µL of 100 µM stock), *dpy-10* repair template (0.5 µL of 10 µM stock), repair template for gene of interest (0.6 µL of 100 µM stock), Cas9 (0.8 µL of 61 µM stock), and water up to 10 µL total. Injection mix components were stored at –20°C, and injection mix was incubated at room temperature for one hour before injections. N2 and MY2147 L4s were picked the day before injecting, and young adults were injected in the gonad and singled to new plates. After 3–4 days, next-generation adults were screened for rollers, which are heterozygous for the *dpy-10* edit and have increased likelihood of also having the desired edit. Non-roller F2 progeny of F1 roller worms were then genotyped to identify worms homozygous for the desired edit, and edits were confirmed by Sanger sequencing. Sequences of sgRNAs, repair templates, and PCR primers for genotyping are available in **Supplementary file 2**.

## Starvation recovery (worm length measurements)

Strains were maintained well-fed for at least three generations prior to beginning experiments. Gravid adults were hypochlorite treated to obtain embryos, which were resuspended at 1 embryo/µL in 5 mL of S-basal in a glass test tube and placed on a roller drum at 20°C so they hatch and enter L1 arrest. After the number of days of L1 arrest indicated on each graph, an aliquot of 500–1000 µL (a consistent volume was used between conditions within the same experiment) per strain was plated on a 10 cm plate with OP50 and allowed to recover for 48 hr. After 48 hr, worms were washed onto an unseeded NGM plate. Images were then taken of worms using a ZeissDiscovery V20 stereomicroscope. To determine lengths of worms, the WormSizer plugin for Fiji was used and worms were manually passed or failed (**Moore et al., 2013**). To determine differences in starvation recovery between strains, a linear mixed-effect model was fit to the length data for all individual worms with duration of starvation and strain as fixed effects and biological replicate as a random effect using the package nlme in R. The summary function was used to calculate a p-value from the t-value.

## Starvation survival

L1 arrest cultures were set up as described for starvation recovery. Starting on the first day of arrest and proceeding every other day, a 100 µL aliquot of culture was pipetted onto a 5 cm NGM plate with a spot of OP50 at the center. The aliquot was placed around the periphery of the lawn, and the

number of worms plated was counted. Two days later, the number of worms that had made it to the bacterial lawn and were alive was counted. Live worms that have developed and were outside the lawn were also counted. The total number of live worms after two days was divided by the total plated to determine the proportion alive. For each replicate, a logistic curve was fit to the data, and the half-life (time at 50% survival) was calculated, and a t-test was performed on half-lives between strains of interest. Power analysis was performed in R using the pwr.t.test function in the 'pwr' package, with parameters n=5, sig.level=0.05, power = 0.5, and type = two.sample. The value of d was calculated, and this was multiplied by the standard deviation of control median survival for that experiment to determine the detectable effect size.

### Early fecundity following starvation

For the assay in *Figure 2G* and L1 arrest cultures were set up as described for starvation recovery. For the assay in *Figure 4C*, the experiment was done in a different lab and worms were arrested in a 15 mL conical tube instead of a glass test tube. Conical tubes were rotated continuously at 20°C. For both figure panels, ~500 L1 larvae were plated on 10 cm OP50 plates at the indicated time point, then allowed to recover for 48 hr. Worms were singled to new 5 cm OP50 plates at approximately 48 hr, then allowed lay progeny until 72 hr, at which time the singled worm was removed. Progeny were counted on these plates 2–3 days later to determine the early fecundity of the individual worms.

### Nuclear localization

L1 arrest cultures were set up as described for starvation recovery. At 36 hr of L1 arrest, an aliquot of 700 µL was spun down in a 1.7 mL Eppendorf tube at 3000 rpm for 30 s to pellet L1 larvae. Of worm pellet, 1.5 µL was pipetted into the center of a slide with a 4% Noble agar pad, and a glass cover slip was immediately placed on top. A timer was set for 3 min, and the slide was systematically scanned with each individual worm scored for nuclear localization at 40 x or 100 x with a Zeiss compound microscope. Nuclear localization of DAF-16::GFP was scored in intestinal cells and assigned as one of four categories: nuclear, more nuclear, more cytoplasmic, and cytoplasmic. 'More nuclear' and 'more cytoplasmic' are intermediate categories between nuclear and cytoplasmic, with localization closer to being nuclear or cytoplasmic, respectively. Scoring for each slide stopped after 3 min. For statistical analysis, nuclear and more nuclear categories were pooled as 'nuclear', while cytoplasmic and more cytoplasmic were pooled as 'cytoplasmic'. The Cochran-Mantel-Haenszel test was used to determine differences in the distribution of the two categories while controlling for biological replicate. See *Figure 4E* for representative images. DAF-16 is initially very nuclear during L1 starvation, and it moves back to the cytoplasm over time during starvation (*Mata-Cabana et al., 2020*). The 36 hr time point was chosen since it is intermediate in this dynamic process.

### Analysis of published RNA-seq data

We analyzed data from three existing publications (*Figure 4—figure supplements 1 and 2*; *Supplementary file 3*; *Cao et al., 2017*; *Webster et al., 2018*; *Taylor et al., 2021*). First, we re-analyzed whole worm bulk mRNA-seq data from fed and starved N2 L1 larvae (four replicates of each condition from a single batch) (*Webster et al., 2018*). Count data was analyzed using edgeR. 60 *irld* genes were part of the protein-coding gene dataset for genome version WS273, and no minimum expression filter was used to restrict the gene set. The calcNormFactors, estimateCommonDisp, and estimateTagwiseDisp functions were used prior to running the exactTest. An FDR cutoff of 0.05 was used to determine significance. For single-cell data across all major worm tissues (*Cao et al., 2017*),Table S4 from the paper was subset to include only *irld* genes, 63 of which were present in the table. Gene expression is represented in *Figure 4—figure supplement 2* when expression levels of transcripts-per-million are at least 1 for that tissue type. For single-cell neuronal data, expression values for *irld* genes were obtained from Supplementary Table 11 of *Taylor et al., 2021*, which includes genes considered expressed at a variety of thresholds and neuronal cell types. Data plotted in *Figure 4—figure supplement 2* uses threshold 3 and data from sensory neurons.

### Materials and correspondence

Correspondence and material requests should be addressed to ryan.baugh@duke.edu.

## Acknowledgements

We thank Oliver Hobert for providing OH16024 *daf-16(ot971[daf-16::GFP])*, Jon Hibshman for sharing a starvation survival curve-fitting script, Chelsea Shoben for help passaging wild isolates, Clay Dilks for CRISPR advice, Sophia Gomez for genotyping assistance, Seth Taylor for strain organization and maintenance, and Jim Jordan for helpful discussions. Funding was provided by the NIH (R01GM117408 and R01GM143159 to LRB and R01ES029930 to ECA and LRB). AKW was supported by an NSF Graduate Research Fellowship. Some strains were provided by the CGC, which is funded by NIH Office of Research Infrastructure Programs (P40 OD010440). We would also like to thank WormBase.

## Additional information

### Funding

| Funder | Grant reference number | Author |
| --- | --- | --- |
| National Institute of General Medical Sciences | R01GM117408 | L Ryan Baugh |
| National Institute of General Medical Sciences | R01GM143159 | L Ryan Baugh |
| National Institute of Environmental Health Sciences | R01ES029930 | Erik C Andersen L Ryan Baugh |

The funders had no role in study design, data collection and interpretation, or the decision to submit the work for publication.

### Author contributions

Amy K Webster, Data curation, Formal analysis, Investigation, Methodology, Supervision, Validation, Visualization, Writing – original draft, Writing – review and editing; Rojin Chitrakar, Data curation, Investigation, Methodology; Maya Powell, Investigation, Validation; Jingxian Chen, Data curation, Formal analysis; Kinsey Fisher, Angela Wei, Validation; Robyn E Tanny, Resources; Lewis Stevens, Kathryn Evans, Formal analysis; Igor Antoshechkin, Data curation, Formal analysis, Investigation, Methodology, Software; Erik C Andersen, Funding acquisition, Resources, Software, Writing – review and editing; L Ryan Baugh, Conceptualization, Funding acquisition, Project administration, Resources, Supervision, Writing – original draft, Writing – review and editing

### Author ORCIDs

Amy K Webster http://orcid.org/0000-0003-4302-8102
Erik C Andersen http://orcid.org/0000-0003-0229-9651
L Ryan Baugh http://orcid.org/0000-0003-2148-5492

### Decision letter and Author response

Decision letter https://doi.org/10.7554/eLife.80204.sa1
Author response https://doi.org/10.7554/eLife.80204.sa2

## Additional files

### Supplementary files

• Supplementary file 1. This file includes all MIP-seq processed data: Slope and PC1 trait values used in GWA, output from MIPgen, MIP sequences, MIPs included in the final experiment, MIP primer sequences, count data for MIP-seq starvation resistance experiment and two pilot experiments.

• Supplementary file 2. This file includes GWA output and follow-up on *irld* candidates: GWA output for both Slope and PC1, genes within QTL, output from WormCat and protein domain enrichment analyses, hyper-divergence status for each strain, CRISPR sequences, genotyping primers, and NIL sequencing results.

- Supplementary file 3. This file includes input and output for RNA-seq analysis, including count tables and differential expression output from *Webster et al., 2018* used in *Figure 4—figure supplement 1*, transcripts-per-million from *Cao et al., 2017* plotted in *Figure 4—figure supplement 2*, and transcripts-per-million for threshold 3 from *Taylor et al., 2021* plotted in *Figure 4—figure supplement 2*.
- Transparent reporting form

## Data availability

Raw MIP-seq data for the starvation-resistance experiment and the pilot experiments to test individual MIPs is available as part of NCBI BioProject PRJNA730178. Code for processing MIP-seq data is available at GitHub (copy archived at swh:1:rev:27839dcc9ef1587086be195349310fb70fbfcaf1). A Source Data file for all figures is also included.

The following dataset was generated:

| Author(s) | Year | Dataset title | Dataset URL | Database and Identifier |
|---|---|---|---|---|
| Webster AK, Baugh LR | 2021 | Population sequencing of C. elegans wild isolates throughout starvation | https://www.ncbi.nlm. nih.gov/bioproject/? term=PRJNA730178 | NCBI BioProject, PRJNA730178 |

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
