## [Editor Report]

The authors identify natural genetic variants in *C. elegans* that are associated with variation in starvation resistance. The authors focus on a gene family (irld's) that are thought to regulate insulin signaling. These studies are very interesting in that the approach for identifying natural gene variants is highly innovative and the work provides novel information about this family of genes.

---

## [Decision Letter]

**Decision letter after peer review:**

[Editors’ note: the authors submitted for reconsideration following the decision after peer review. What follows is the decision letter after the first round of review.]

Thank you for submitting the paper "Natural variation in the *irld* gene family affects insulin/IGF signaling and starvation resistance" for consideration by *eLife*. Your article has been reviewed by 3 peer reviewers, and the evaluation has been overseen by a Reviewing Editor and a Senior Editor. The following individuals involved in review of your submission have agreed to reveal their identity: Patrick T McGrath (Reviewer #3).

We are sorry to say that, after consultation with the reviewers, we have decided that this work is not currently suitable for publication in *eLife*. All three reviewers expressed an overall interest in the potential scope and importance of the work. However, as you can see detailed in the reviews detailed below, the reviewers questioned the strength of the evidence that implicated the fascinating ilrd genes in insulin receptor signaling. This concern was further amplified in an extensive and robust discussion that the 3 reviewers had after seeing each other's reviews. For example, one reviewer pointed out that manipulations in TGF-β signaling have similar DAF-16-related read-outs that you describe here for ildd gene manipulation; hence, these read-outs are not sufficient proof for direct involvement of these genes in insulin receptor signaling.

While we all agree that the paper is presently not a candidate for publication in *eLife*, we would be interested in seeing a very substantially revised version of this manuscript in which the function of the ilrd genes is more precisely delineated.

*Reviewer #1:*

In this manuscript, the authors interrogate a large panel of wild *C. elegans* strains to identify natural genetic variants that influence starvation resistance. They use molecular inversion probe sequencing (MIP-Seq) to rapidly identify specific strains in a pool of wild strains that are resistant or sensitive to starvation. By taking advantage of the *C. elegans* Natural Diversity Resource, they perform genome-wide association studies to identify quantitative trait loci (QTL) that influence starvation resistance. They validate these QTLs by constructing near-isogenic lines. Detailed analysis of these QTLs reveals variants in irld genes that are shown to influence organismal growth after starvation recovery. irld genes are hypothesized to encode extracellular proteins that may bind to insulin-like growth factors. Based on functional analysis of variants in irld-39 and irld-52, the authors propose a model in which IRLD-39 and IRLD-52 influence starvation resistance by modulating signaling through the insulin receptor homolog DAF-2.

The major strength of this study is the identification of natural genetic variants that influence starvation resistance. The authors use a creative and powerful approach that in principle can be used in any organism to elucidate the genetic architecture of any phenotypic trait. This aspect of the manuscript will be of general interest.

In my opinion there are four major weaknesses of the manuscript. First, the authors use organismal length after recovery from starvation as a surrogate phenotype for starvation resistance. I am not convinced that this is justified, as the post-recovery organismal length of one of the starvation-sensitive strains identified in the study is not significantly different from that of the two most starvation-resistant strains identified (Figure 2F). Additionally, insufficient information and characterization of the irld-39/52 variants is provided. If these are non-coding variants, it would be premature to conclude that they affect irld-39/52 function without supporting data. The functional analysis of the irld-39 and irld-52 variants does not convincingly support the authors' model of IRLD-39/52 acting through the DAF-2 insulin-like pathway. Related to this point, no experiments are presented to test the possibility that these variants influence LET-23/EGFR signaling, although IRLD proteins are reported to have homology to EGF receptors as well as insulin receptors.

1. Lines 100-102: What is the nature of the irld-39 and irld-52 variants? Are they intronic or exonic? If they are non-coding, then data is needed to show that they influence irld-39/52. Are they loss- or gain-of-function, and why? Why are they "high-impact"?

2. Lines 108-109: Post-starvation length is a direct measure of growth in response to refeeding. Here it is being used as a surrogate measure of starvation recovery. How is "recovery" defined? One could use post-recovery survival as a measure of "recovery," but I can imagine that post-recovery fecundity might be a better measure of recovery from an evolutionary standpoint. If the authors are going to use organismal length as a surrogate phenotype, they need to show that this phenotype tracks with a more biologically relevant "recovery" phenotype. The data for NIC526 (Figures 2E-F) suggest that post-recovery length may not be a good indicator of starvation resistance.

3. Figures 3J-K and 4A-B: Starvation resistance assays should be performed on these strains (e.g. Figure 2E).

4. Figure 4C: The DAF-16 localization data are not convincing. The results show a modest difference, the biological significance of which is unclear. Was the experimenter blinded to the identity of the strain being observed? How was the 36-hour time point chosen, and why is this more biologically relevant than other time points?

5. Line 139: Based on the Methods section, it appears that the authors are using daf-2(e1370), which is a strong lof allele. I don't think this is the right allele to use in these studies; DAF-16 is so strongly activated in daf-2(e1370) compared to the modest effect of irld-39;irld-52 on DAF-16 localization (Figure 4C) that it could easily obscure subtler effects of irld-39/52 on gene expression, regardless of whether DAF-2 acts downstream of or parallel to IRLD-39/52.

6. Line 146: The fact that DAF-16 target gene expression "reverses later in starvation" contradicts the authors' model. This observation warrants further experimentation.

7. The key transcriptome experiments to test the authors' model are missing. They need to show that changes in gene expression caused by manipulation of irld-39 and irld-52 activity are DAF-16-dependent.

*Reviewer #2:*

In this study, Webster et al. have aimed to identify the genetic factors that contribute to the differences between different wild *C. elegans* strains in terms of their resistance to starvation. The genomic sequences of hundreds of wild *C. elegans* strains have become recently available and this has given the opportunity to investigate the genetic determinants of the physiological differences between these wild populations that were isolated from different ecological niches. Here, the authors have subjected a mixture of wild *C. elegans* strains to long periods of starvation during early larval development and have utilized genomic sequencing to quantify the relative enrichment of each individual wild strain after exposure to starvation for different time intervals. Using the genomic sequencing strategy called MIP-Seq, they have identified two wild *C. elegans* strains that are overrepresented in the mixed population after extended starvation (implying higher starvation resistance compared to other wild strains) and they have also found two wild strains that are underrepresented after extended starvation (implying lower starvation resistance compared to other wild strains).

Using genome-wide association (GWA) analyses for parameters of starvation resistance, they have identified quantitative trait loci (QTL) associated with this phenotype. The genes enriched in these QTLs include multiple members of the insulin/EGF-receptor L domain (IRLD) gene family. The irld genes encode proteins that have extracellular ligand binding domains, but no receptor tyrosine kinase domains, and their function remains largely unknown. By introducing allelic variants for irld genes from the stress-resistant wild strains in the genetic background of the stress-sensitive strains using Crispr, the authors were able to improve the stress resistance of these sensitive strains, thus showing a direct role of irld genes in starvation resistance. Using epistasis experiments, they further demonstrate that irld genes might improve survival via interacting with insulin/IGF signaling in worms. Based on recently published neuronal single-cell RNA-seq data, the authors propose that irld genes function in specific sensory neurons to control starvation resistance in animals. How IRLD proteins modulate insulin signaling in a small subset of neurons to affect an organism-level phenotype and the underlying mechanism that likely involves interorgan signaling remain elusive.

There are several strengths of this study such as:

(1) The irld gene family has undergone large expansion in nematodes but their biological function remains mostly unknown. This study provides the first evidence for the role of this gene family in starvation resistance and thus indicating that the expansion of irld genes in nematodes might be a major contributing factor to the success of nematode species in colonizing a wide range of ecological niches.

(2) Through the innovative use of MIP-Seq, the authors have laid the foundation for a quantitative approach to measure differences in complex physiological traits in a mixed population of individuals that are genetically heterogenous. The same strategy can be useful for many other experimental paradigms such as studying the differences in resistance to physiological stressors, non-uniform effects of pharmacological compounds or variability in normal aging in genetically heterogenous wild populations.

(3) The irld mutants and the associated RNA-seq datasets generated in this study will be invaluable for *C. elegans* researchers to further investigate the potential roles of this understudied family of genes in regulating physiology, behavior and metabolism of animals.

However, the claims made in this study have limitations and shortcomings that are primarily attributable to the use of some suboptimal experimental strategies, which are listed below:

(1) To identify genes in the QTL for starvation resistance, the authors have looked for enrichment of gene symbol prefixes. Though they have identified the irld genes, which they demonstrate to be functionally related to starvation resistance, this approach is suboptimal because gene symbol prefixes in *C. elegans* are not always representative of gene function, but instead they have historically represented the phenotype of the mutant (e.g. 'let' for lethal, 'eat' for abnormal eating, 'unc' for uncoordinated etc.). Hence not all genes with the same gene symbol prefix have related biological functions, neither do all genes of the same gene family have the same gene symbol prefix. Hence, it is likely that the authors have missed out on identifying all the gene families that are enriched in the QTL for starvation resistance.

(2) The central message of the paper is that the irld genes regulate starvation resistance. There are two key components of this phenotype: (a) survival during starvation, and (b) recovery from starvation. In their initial experimental strategy to validate the starvation resistance of wild strains identified from MIP-Seq, the authors have performed assays for both starvation survival (proportion of surviving worms at different time points of starvation) and starvation recovery (body length measurement post 48 hr of recovery from starvation). However, all their subsequent analyses involving irld gene manipulations and interactions with insulin signaling only utilized the body length measurement assay. Since body length measurement is only a measure of starvation recovery but not of starvation survival, it is not possible to conclude whether irld gene manipulations improve overall survival of animals during starvation. Hence, a key measure of starvation resistance is missing from the methodology that has been used here to study the effect of irld genes.

(3) The claim that irld genes are predominantly expressed in sensory neurons is made using a dataset that did not have the expression profiles for non-neuronal tissues. Since tissues such as intestine, muscles and hypodermis have important roles in dictating organism-level phenotypes, it is essential to know whether the irld genes are expressed in these tissues. The tissue-restricted role of irld genes in starvation resistance that is proposed in the study can be addressed if the cell type-specific expression pattern of irld genes during normal and starvation conditions is known.

– Figure 3D: For this gene enrichment analysis, genes with the same symbol prefix were considered as part of the same gene family (line 362 in Methods). However, gene symbol prefixes in *C. elegans* are not always representative of gene function (e.g. let, eat, unc etc.). Hence, searching for enrichment of gene symbol prefixes might lead to misleading and incomplete results. For example, not all of the 283 genes with the 'nhr' gene symbol prefix (that the authors report in Figure 3D) belong to the NHR gene family. Many of these 'nhr' genes are pseudogenes (nhr-75, nhr-83, nhr-220 etc.) and many other genes with the 'nhr' symbol do not have the C4-zinc finger DNA binding domain that is a characteristic of members of the NHR gene family. Furthermore, not all members of the NHR gene family have 'nhr' gene symbol prefixes (e.g. daf-12, dpr-1, odr-7, unc-55 etc.). Instead of looking for enrichment of gene symbol prefixes, the authors should search for enrichment of specific protein domains (InterPro, Pfam etc.). This might reveal enrichment of genes belonging to other functional categories, in addition to the irld gene family identified here.

– Figures 3J, 3K, 4A, 4B: Measuring the body length of worms after 48 hr of recovery from starvation should not be the only parameter to quantify the starvation resistance of a particular genotype. The authors should also perform the standard starvation survival assays (similar to data shown in Figure 2E) for irld gene manipulations in N2 and MY2147 genetic backgrounds and also for the strains in which the interaction of irld genes with insulin signaling was investigated.

– Line 152: That authors should clarify that irld gene expression is restricted to sensory neurons among neuron types. The single-cell RNA-seq dataset they have utilized here reports expression only among neuron classes, but not in non-neuronal tissues. Firstly, the authors should repeat this analysis with the unthresholded dataset from Taylor et al. and remake figures 4H and figure S6 using expression data from both neuronal and non-neuronal tissues. Since non-neuronal tissues such as the intestine, muscles and hypodermis likely have important roles in determining starvation resistance of the animals, it is crucial to look at the expression of irld genes in these non-neuronal tissues as well. Secondly, the single-cell RNA-seq strategy in Taylor et al. was primarily designed to identify gene expression in neurons. Hence, the gene expression data from non-neuronal tissues might have not detected medium or weak expression of genes in non-neuronal tissues. Since the authors make the claim that irld-39 and irld-52 function primarily in sensory neurons to affect starvation resistance (line 159), it would be prudent to make Crispr-based transcriptional GFP reporters for these two genes and validate whether their expression is indeed restricted to ASJ and ADL neurons, respectively. These reporters should also be used to demonstrate whether the expression of irld genes changes in these neurons during exposure to starvation. If no expression is detected in non-neuronal tissues, this would strengthen the argument that irld genes are expressed in an anatomically restricted manner only in specific sensory neurons, but they regulate starvation resistance at the organism level potentially via systemic signaling. This would signify presence of cell non-autonomous effects of irld genes, which can be investigated in future studies.

*Reviewer #3:*

The authors make two major findings. First, they adapt MIP-seq to *C. elegans*, identifying primers that can identify individual strains and quantitate their relative proportion in group competition experiments. Second, they use this technique to map loci responsible for natural variation in starvation response, identifying irld family member genes that influence survival to starvation.

The development of MIP-seq is a major achievement. Other labs can use the primers that they develop to perform similar experiments, competing wild strains against each other in their assay of interest. Because the wild strains are already sequenced, GWAS can be performed without the cost of any sequencing, using available software that is provided by the authors. One potential issue with the adaptability of this approach is the potential for outcrossing during the competition phase. While this is not an issue for the starvation mapping they perform here, it will need to be addressed to make this technique generalizable. However, enthusiasm for this approach remains high; competition in large group settings provides a much better handle on fitness than more common assays.

Besides mapping and validating the loci that are responsible for variation in starvation response, they also identify causal mutations in irld genes. Using CRISPR-Cas9, they demonstrate a role for two natural mutations in starvation response and also implicate two additional irld genes as well. Use of CRISPR-Cas9 to specifically edit the genome are the gold standard for demonstrating a causal role for specific mutations. irld genes are homologous to insulin/EGF receptor proteins and this work implicates insulin signaling in starvation response. Additionally, they use classical genetics to implicate insulin signaling using epistasis experiments with the FOXO DAX-16 transcription factor. In general, starvation is poorly understood in humans and other species. This provides important evidence that insulin might be involved.

1. It is interesting that these genes are primarily expressed in sensory neurons. It is probably useful to use rescue experiments to show that this is the case.

2. Since the strains exists, some experiments on the CRISPR/Cas9 allelic-replacement strains in normal well-fed conditions would be useful. Do these strains have the same lifespan, store the same amount of fat, and eat the same amount of food during non-starvation conditions that could help explain why they survive longer?

3. There is little discussion about the generalizability of this work to other species. What is known or thought about variation in insulin pathways and starvation? Are there specific example in other species? How does this affect how we think about starvation and natural variation in starvation in humans and other species?

[Editors’ note: further revisions were suggested prior to acceptance, as described below.]

Thank you for resubmitting your work entitled "Natural variation in the *irld* gene family affects starvation resistance in *C. elegans*" for further consideration by *eLife*. Your revised article has been evaluated by David James (Senior Editor) and a Reviewing Editor.

The manuscript has been improved but there are some remaining issues that need to be addressed, as outlined below:

The reviewers have discussed their reviews with one another quite extensively. As you can see from their initial set of comments below, there still remains a substantial concern among all reviewers about whether and to what extent irld genes are involved in the starvation response. However, all reviewers have appreciated your creative new use of the MIP-Seq technology which we all expect to be quite impactful for further studies in *C. elegans*. We recommend that you (a) shorten the paper to a Report format, and (b) focus the paper on the technology aspect and provide the irld genes as an application. The present abstract of your paper organizationally already hints toward such a format, but this strategy should be implemented for the rest of the paper as well (including the Introduction). Also, as indicated by all of the reviewer's comments, please revise the manuscript editorially to consider alternative explanations of your data since, as you can see, the reviewers remain unconvinced that irld genes have been strongly implicated in this phenomenon.

*Reviewer #1:*

In the revised format, the manuscript is coherent and concise. This study has used MIP-seq to identify that genetic variation in the irld gene family determines some aspects of starvation resistance in wild worm isolates. However, the manuscript lacks any evidence regarding the mechanism and site of action of irld genes except the finding that one of the three phenotypes is DAF-16-dependent. Given the limited depth and breadth of this study, it is more suitable for the Short Reports format rather than the Research Article format.

Regarding the response to major comment #2 (Reviewer 2), the authors have not measured the starvation resistance of the irld-39; irld-52 double mutant in the MY2147 background. They quantified these phenotypes only in the N2 background, where the effects are either modest or not significant. Since single mutant manipulations produce much stronger phenotypes in the MY2147 background compared to N2 (Figure 3K), it is likely that the double mutant might display a robust increase in starvation resistance in the stress-sensitive MY2147 background. This experiment, though not essential, will greatly increase the impact of the study and will indicate that simultaneous manipulation of only a handful of irld genes can completely ameliorate the high stress-sensitivity in a wild strain.

*Reviewer #2:*

I appreciate the efforts that the authors have undertaken to address my critique of the initial submission, and I also commend them for their transparency about their results. However, I remain unconvinced about two of the authors' claims: that the irld variants they focus on account for the differences in starvation resistance observed in wild strains, and that irld-39/52 act through DAF-16 to modulate starvation resistance.

1. Figures 3I/J: this data shows that the irld edited strains do not confer improved starvation survival on the sensitive MY2147 background. While the authors offer the interpretation that "…the variants primarily affect starvation recovery," another more parsimonious explanation would be that these variants are not the key functional variants within the QTLs identified using MIP-seq.

2. The authors mention that "…the irld double mutant did not cause statistically significant changes in the expression of individual genes." Their explanation for this observation is that "We believe there must be differences in gene expression, but that they are relatively small and in specific tissues, thus obscured by analyzing whole worms. We also believe this is a testament to the sensitivity of our phenotypic assays." To me, the most parsimonious explanation for the observation that DAF-16 target gene expression is not influenced by irld-39/52 mutation is that the small increase in DAF-16 nuclear localization observed is not functionally significant. Moreover, while their phenotypic assays may be sensitive, the other more straightforward (IMO) explanation is that their phenotypic assays are capturing effects of other variants within the identified QTLs that are distinct from the irld variants that the authors have chosen to focus on.

3. The QTLs identified by the authors range from >600kb to >2.2Mb by my estimate. How many polymorphisms lie within these intervals? I understand the interest in the irld gene family, but the data do not convince me that the irld variants in question are the key functional variants within these intervals.

*Reviewer #3:*

The authors utilize a standard starvation assay to study natural variation in starvation response among wild strains of *C. elegans*. Taking advantage of the CeNDR database, the authors compete ~100 wild strains against each other and quantify the changes in population using MIP-seq. The authors quite convincingly show that differences in survival occur among the wild strains and use GWAS to non-biasedly identify regions of the genome that are associated with differences in survival in this paradigm (QTLs). Technically this is quite challenging, and the identification of the QTLs was verified nicely using near isogenic lines.

Throughout the paper, the authors describe the differences in survival in these conditions as starvation resistance. However, additional factors could be at play, such as differential susceptibility to toxins or pheromones that could build up during the multiday experiment.

By analyzing the QTLs, the authors identified a family of genes, irlds, which were enriched within these regions. These genes are upregulated by starvation and have homology to insulin-type receptors. The authors propose that natural variation in these genes is important for natural variation in starvation response. Two demonstrate this, the authors identify two irld genes that carry likely loss of function deletions and use CRISPR/Cas9 to engineer these mutations into other genetic backgrounds. Additionally, the authors' engineer deletion alleles (that do not mimic segregating alleles) in two additional irld genes.

Using these alleles, the authors convincingly show that irld genes play a role in this starvation assay, demonstrating differences in growth during and after exit from starvation. This result is likely to be exciting to researchers interested in starvation, as insulin signaling is an important genetic pathway in a large number of organisms.

While the authors also interpret their data to conclude that natural genetic variation in irld genes is important for natural variation in starvation resistance, I am less convinced by this data. (1) For the artificial deletion alleles of irld-57 and irld-11, functional differences in their protein activity or expression is not presented. How do the authors know that genetic variation among wild strains leads to functional differences in these proteins? Additionally, quantitative complementation (or a similar approach) was not performed to demonstrate that differences in their function exist between different wild strains. (2) For the putative lof allele in irld-39, no data is shown to demonstrate that this affects IRLD-39 activity. This 5bp deletion is also close to the 5' end of a nearby gene – could the deletion be affecting the expression of this other gene? (3) The deletion of irld-52 is probably the most convincing, however, again, no evidence supporting its role as a loss of function allele is presented beyond sequence analysis such as isolation of cDNAs. Was this entire region sequenced to verify its existence and its predicted effect in wild strains (i.e. are there other genetic variants that might suppress the frameshift nature of this deletion?).

The authors also often switch back and forth between assays, which also makes it difficult to interpret the importance of the natural genetic variants in the overall differences in wild strains. Sometimes linear models are used to analyze strains (slope and intercept), sometimes size during starvation is used, and sometimes recovery is used. Because the NILs were not used as controls for many of these experiments, it is impossible to compare the effect of the individual mutations to the effect of the locus that has been mapped. Do these lof alleles represent the majority of the effect of this locus, or is this a minor effect and other much more important alleles remain to be found.

1. Either additional analysis to make the claim that the differences in survival is a starvation response or changes to the text to discuss alternative possibilities.

2. Further analysis to demonstrate that the natural 5bp deletions cause functional differences in IRLD protein

3. Additional experiments or inclusion of existing data that allow the comparison of the effect size of the CRISP'ed strains to the appropriate NIL for comparison.

---

## [Author Response]

[Editors’ note: the authors resubmitted a revised version of the paper for consideration. What follows is the authors’ response to the first round of review.]

Reviewer #1:In this manuscript, the authors interrogate a large panel of wild *C. elegans* strains to identify natural genetic variants that influence starvation resistance. They use molecular inversion probe sequencing (MIP-Seq) to rapidly identify specific strains in a pool of wild strains that are resistant or sensitive to starvation. By taking advantage of the *C. elegans* Natural Diversity Resource, they perform genome-wide association studies to identify quantitative trait loci (QTL) that influence starvation resistance. They validate these QTLs by constructing near-isogenic lines. Detailed analysis of these QTLs reveals variants in irld genes that are shown to influence organismal growth after starvation recovery. irld genes are hypothesized to encode extracellular proteins that may bind to insulin-like growth factors. Based on functional analysis of variants in irld-39 and irld-52, the authors propose a model in which IRLD-39 and IRLD-52 influence starvation resistance by modulating signaling through the insulin receptor homolog DAF-2.The major strength of this study is the identification of natural genetic variants that influence starvation resistance. The authors use a creative and powerful approach that in principle can be used in any organism to elucidate the genetic architecture of any phenotypic trait. This aspect of the manuscript will be of general interest.

Thank you for identifying the major strengths of our manuscript. We believe the revised manuscript makes those strengths more accessible by more clearly presenting the work.

In my opinion there are four major weaknesses of the manuscript. First, the authors use organismal length after recovery from starvation as a surrogate phenotype for starvation resistance. I am not convinced that this is justified, as the post-recovery organismal length of one of the starvation-sensitive strains identified in the study is not significantly different from that of the two most starvation-resistant strains identified (Figure 2F). Additionally, insufficient information and characterization of the irld-39/52 variants is provided. If these are non-coding variants, it would be premature to conclude that they affect irld-39/52 function without supporting data. The functional analysis of the irld-39 and irld-52 variants does not convincingly support the authors' model of IRLD-39/52 acting through the DAF-2 insulin-like pathway. Related to this point, no experiments are presented to test the possibility that these variants influence LET-23/EGFR signaling, although IRLD proteins are reported to have homology to EGF receptors as well as insulin receptors.

We agree that the four weaknesses highlighted warranted further explanation and/or experimentation. First, we have expanded our explanation of starvation resistance in the introduction. In brief, starvation resistance includes survival during starvation, recovery following starvation, and fecundity following starvation. We have used this more inclusive perspective on starvation resistance in multiple publications (Jobson
*et al.* 2015; Hibshman
*et al.* 2016; Webster
*et al.* 2018; Jordan
*et al.* 2019; Webster
*et al.* 2019; Chen
*et al.* 2022), and it has been described with extensive citations in a WormBook chapter on starvation (Baugh and Hu 2020). Our MIP-seq experimental design includes all three of these facets of starvation resistance, because a strain could be considered relatively starvation resistant even if it survives the same as another strain, as long as it recovers faster. This is now explained in the Results section. Our experiments to follow up on MIP-seq were thus designed to determine which aspect(s) of starvation resistance rendered a given strain more or less resistant in the MIP-seq experiment. We have added starvation survival results throughout, and in some cases we have also added data for early fecundity following starvation. In cases where we rely on only one of the three starvation-resistance assays we explain our rationale.

As pointed out by the reviewer, one of the sensitive strains recovered well (despite displaying reduced survival and early fecundity) suggesting this aspect of starvation resistance was not the driving force behind its sensitivity. There are documented cases of these phenotypes being coupled or de-coupled, some of which are now cited in the Introduction, but all are of importance.

Second, we have now further documented the exact *irld* variants investigated in the text and Figure 3. This information for all variants within the QTL is also still available as part of the supplementary data. We specifically chose variants that would affect the coding sequence of the gene. This is now made clear in Figure 3 and the Results section. We spell out the predicted effects of each variant on gene function, why we chose those variants for further investigation, and the rationale for our genome-editing strategy in each case.

Third, we have pulled back on our interpretation involving DAF-2/InsR. Our data show that *irld* function is dependent on DAF-16/FoxO in the context of starvation recovery, and DAF-16 localization is affected during starvation. However, we do not explicitly show involvement of DAF-2. We have softened our conclusions on this throughout, including the title, Abstract, sub-headers, figure titles, and Discussion. However, we speculate about IRLD function in the Discussion, putting forward the suggestion that IRLD proteins function in part through modification of insulin/IGF signaling, as originally proposed based on homology (Dlakic 2002). We believe the reader deserves to hear our thoughts on this, and this is the best model to account for our results. We recognize that we have not, for example, demonstrated that the IRLD proteins directly bind insulin-like peptides (ILPs). However, it is worth noting that none of the 40 *C. elegans* ILPs have been shown to actually bind the insulin/IGF receptor protein DAF-2. Furthermore, a similar model has been proposed and recently published in *eLife* regarding the truncated *daf-2* isoform *daf-2B*, also without biochemical support (Martinez
*et al.* 2020). We also do not think our hypothetical model is outlandish given a similar model for the function of IGF-binding proteins.

We also agree that LET-23/EGFR signaling could be involved, as *hpa-1* and *hpa-2* have been shown to interact with EGFR signaling. While we did not claim EGFR signaling was not involved, we agree that focusing on insulin/IGF signaling could leave the reader with the impression that it is the primary regulator. We now explicitly point out that EGF signaling could be modified by IRLD function, and we explicitly reference the *hpa-1/hpa-2* paper in the Introduction and Discussion. However, it should be noted that EGF has not been investigated in the context of starvation or L1 arrest, unlike insulin/IGF signaling, and so it would be a significant undertaking to address this possibility, which we believe is beyond the scope of this manuscript.

1. Lines 100-102: What is the nature of the irld-39 and irld-52 variants? Are they intronic or exonic? If they are non-coding, then data is needed to show that they influence irld-39/52. Are they loss- or gain-of-function, and why? Why are they "high-impact"?

Thank you for this very important point. We have added an explanation of why both variants were chosen. Both are loss-of-function variants predicted to disrupt protein function. *irld-39* disrupts the start codon, likely rendering the gene a null in starvation resistant strains. *irld-52* contains a variant predicted to disrupt the fifth exon, which is likely loss-of-function, but it is unclear if it is a null. Figure 3 and the Results now spell out the nature of each variant.

2. Lines 108-109: Post-starvation length is a direct measure of growth in response to refeeding. Here it is being used as a surrogate measure of starvation recovery. How is "recovery" defined? One could use post-recovery survival as a measure of "recovery," but I can imagine that post-recovery fecundity might be a better measure of recovery from an evolutionary standpoint. If the authors are going to use organismal length as a surrogate phenotype, they need to show that this phenotype tracks with a more biologically relevant "recovery" phenotype. The data for NIC526 (Figures 2E-F) suggest that post-recovery length may not be a good indicator of starvation resistance.

Thank you for raising this very important point as well. We have now expanded our explanation of starvation resistance and rationale for the various assays used in the text. In particular, we note that the MIP-seq starvation resistance experiment incorporates aspects of survival, recovery, and fecundity in the design, so a given strain may be starvation resistant or sensitive due to any of these individual phenotypes. We believe this is the best proxy for fitness, which presumably has multiple parameters. As pointed out, NIC526 recovers relatively well following starvation, but does not survive well, suggesting that survival may drive its sensitivity in the MIP-seq assay. We have cited literature showing that these aspects of starvation resistance can be decoupled (though it is not uncommon that they are well correlated). Rather than delegitimize the use of starvation recovery as an assay, NIC526 further highlights the relevance of assaying multiple phenotypes. We agree that looking at early fecundity is also important, given relevance to fitness. We have now performed this experiment for key strains. We specifically performed early fecundity assays for DL238, EG4725, MY2147, and NIC526, which showed that DL238 and EG4725 are starvation resistant relative to MY2147 and NIC526. Because MY2147 is sensitive in all three assays, it was a strong candidate for use in NIL generation and genome editing. We also now show that *irld-39(duk1); irld-52(duk17)* exhibits increased fecundity following starvation compared to the N2 control. These results are critical in that they demonstrate that early fecundity is affected in the most resistant and sensitive strains identified as well as the double mutant we analyze in the N2 background.

3. Figures 3J-K and 4A-B: Starvation resistance assays should be performed on these strains (e.g. Figure 2E).

While we note that starvation recovery is a starvation-resistance assay, we have added starvation survival results for all of these strains as well. Survival results were not statistically significant, suggesting that the variants primarily affect starvation recovery. We have also added results from a power analysis so that we can state how large of an effect size would have been needed to have obtained statistical significance.

4. Figure 4C: The DAF-16 localization data are not convincing. The results show a modest difference, the biological significance of which is unclear. Was the experimenter blinded to the identity of the strain being observed? How was the 36-hour time point chosen, and why is this more biologically relevant than other time points?

We are sorry that this presentation was not more convincing. As another reviewer pointed out, the pictures made it difficult to tell the difference between different categories and were small. We have therefore generated new, enlarged images. We have also binarized the data to better represent the difference between the categories. While this assay alone does not show the biological significance of DAF-16 localization, the starvation resistance assay, showing that increased starvation resistance of worms with *irld* variants depends on DAF-16, suggests biological importance. DAF-16 is initially very nuclear during L1 starvation, and it moves back to the cytoplasm over time during starvation (Mata-Cabana
*et al.* 2020). The 36-hour time point was chosen since it is intermediate in this dynamic process. This is now explained in the methods section.

5. Line 139: Based on the Methods section, it appears that the authors are using daf-2(e1370), which is a strong lof allele. I don't think this is the right allele to use in these studies; DAF-16 is so strongly activated in daf-2(e1370) compared to the modest effect of irld-39;irld-52 on DAF-16 localization (Figure 4C) that it could easily obscure subtler effects of irld-39/52 on gene expression, regardless of whether DAF-2 acts downstream of or parallel to IRLD-39/52.

Although we used *daf-2(e1370)* at 20ºC, rendering it not as strong of an allele, we have opted to remove this experiment since it is subject to various interpretations.

6. Line 146: The fact that DAF-16 target gene expression "reverses later in starvation" contradicts the authors' model. This observation warrants further experimentation.

Yes, this result does give one pause. But we actually don't believe that it contradicts our model, but instead that it is indicative of the complexity of insulin signaling dynamics in this multicellular system with agonists, antagonists, feedback, etc. However, we recognize that this *ad hoc* explanation is not satisfying, and we have opted to remove the gene expression analysis.

7. The key transcriptome experiments to test the authors' model are missing. They need to show that changes in gene expression caused by manipulation of irld-39 and irld-52 activity are DAF-16-dependent.

We agree that this is a very intriguing experiment, especially since we show that the increase in starvation resistance caused by disruption of *irld-39* and *irld-52* depends on *daf-16.* However, the *irld* double mutant did not cause statistically significant changes in the expression of individual genes, thus our previous analyses of *daf-16* targets as a group and transcriptome-wide epistasis with *daf-2*, both of which we have removed. Without differentially expressed genes, we do not see a robust way to analyze such an RNA-seq epistasis analysis of *daf-16* and *irld-39; irld-52*, and we therefore decided not to include it*.* By the way, it may be considered troubling that we report phenotypic effects of this double mutant with no differentially expressed genes detected. This is not the first time we have encountered this phenomenon (Webster
*et al.* 2018). We believe there must be differences in gene expression, but that they are relatively small and in specific tissues, thus obscured by analyzing whole worms. We also believe this is a testament to the sensitivity of our phenotypic assays.

Reviewer #2 (Recommendations for the authors):In this study, Webster et al. have aimed to identify the genetic factors that contribute to the differences between different wild *C. elegans* strains in terms of their resistance to starvation. The genomic sequences of hundreds of wild *C. elegans* strains have become recently available and this has given the opportunity to investigate the genetic determinants of the physiological differences between these wild populations that were isolated from different ecological niches. Here, the authors have subjected a mixture of wild *C. elegans* strains to long periods of starvation during early larval development and have utilized genomic sequencing to quantify the relative enrichment of each individual wild strain after exposure to starvation for different time intervals. Using the genomic sequencing strategy called MIP-Seq, they have identified two wild *C. elegans* strains that are overrepresented in the mixed population after extended starvation (implying higher starvation resistance compared to other wild strains) and they have also found two wild strains that are underrepresented after extended starvation (implying lower starvation resistance compared to other wild strains).Using genome-wide association (GWA) analyses for parameters of starvation resistance, they have identified quantitative trait loci (QTL) associated with this phenotype. The genes enriched in these QTLs include multiple members of the insulin/EGF-receptor L domain (IRLD) gene family. The irld genes encode proteins that have extracellular ligand binding domains, but no receptor tyrosine kinase domains, and their function remains largely unknown. By introducing allelic variants for irld genes from the stress-resistant wild strains in the genetic background of the stress-sensitive strains using Crispr, the authors were able to improve the stress resistance of these sensitive strains, thus showing a direct role of irld genes in starvation resistance. Using epistasis experiments, they further demonstrate that irld genes might improve survival via interacting with insulin/IGF signaling in worms. Based on recently published neuronal single-cell RNA-seq data, the authors propose that irld genes function in specific sensory neurons to control starvation resistance in animals. How IRLD proteins modulate insulin signaling in a small subset of neurons to affect an organism-level phenotype and the underlying mechanism that likely involves interorgan signaling remain elusive.There are several strengths of this study such as:(1) The irld gene family has undergone large expansion in nematodes but their biological function remains mostly unknown. This study provides the first evidence for the role of this gene family in starvation resistance and thus indicating that the expansion of irld genes in nematodes might be a major contributing factor to the success of nematode species in colonizing a wide range of ecological niches.(2) Through the innovative use of MIP-Seq, the authors have laid the foundation for a quantitative approach to measure differences in complex physiological traits in a mixed population of individuals that are genetically heterogenous. The same strategy can be useful for many other experimental paradigms such as studying the differences in resistance to physiological stressors, non-uniform effects of pharmacological compounds or variability in normal aging in genetically heterogenous wild populations.(3) The irld mutants and the associated RNA-seq datasets generated in this study will be invaluable for C. elegans researchers to further investigate the potential roles of this understudied family of genes in regulating physiology, behavior and metabolism of animals.

Thank you for pointing out strengths of our study.

However, the claims made in this study have limitations and shortcomings that are primarily attributable to the use of some suboptimal experimental strategies, which are listed below:(1) To identify genes in the QTL for starvation resistance, the authors have looked for enrichment of gene symbol prefixes. Though they have identified the irld genes, which they demonstrate to be functionally related to starvation resistance, this approach is suboptimal because gene symbol prefixes in *C. elegans* are not always representative of gene function, but instead they have historically represented the phenotype of the mutant (e.g. 'let' for lethal, 'eat' for abnormal eating, 'unc' for uncoordinated etc.). Hence not all genes with the same gene symbol prefix have related biological functions, neither do all genes of the same gene family have the same gene symbol prefix. Hence, it is likely that the authors have missed out on identifying all the gene families that are enriched in the QTL for starvation resistance.

Thank you for pointing this out. We agree that using gene prefixes is not the ideal approach. We thus performed a new analysis to replace this panel, the details of which are in the second paragraph of the ‘Enrichment analyses’ section. In brief, we used a fasta file of protein sequences of the whole genome as the background set, and a fasta file of protein sequences for the set of genes with variants in QTL to determine protein domain enrichments. We used the hmmscan program to search the Pfam database for protein domains from each set and then calculated a hypergeometric p-value of overlap to determine significance. The receptor L domain is significantly enriched, and this is the domain that defines *irld* genes.

(2) The central message of the paper is that the irld genes regulate starvation resistance. There are two key components of this phenotype: (a) survival during starvation, and (b) recovery from starvation. In their initial experimental strategy to validate the starvation resistance of wild strains identified from MIP-Seq, the authors have performed assays for both starvation survival (proportion of surviving worms at different time points of starvation) and starvation recovery (body length measurement post 48 hr of recovery from starvation). However, all their subsequent analyses involving irld gene manipulations and interactions with insulin signaling only utilized the body length measurement assay. Since body length measurement is only a measure of starvation recovery but not of starvation survival, it is not possible to conclude whether irld gene manipulations improve overall survival of animals during starvation. Hence, a key measure of starvation resistance is missing from the methodology that has been used here to study the effect of irld genes.

It is clear that this is a very important point, since it has been raised by multiple reviewers. Please see the above comments on this point. In brief, starvation survival as well as growth and fecundity following starvation are all important aspects of starvation resistance. A strain may be more or less resistant due to differences in any of these phenotypes, and though they are often correlated, there are also documented examples of them being decoupled. This is now explained in the Introduction. We have also made a point of explaining in the Results that our MIP-seq sample collection integrates effects on all three of these phenotypes, which we believe is the best proxy for fitness. And we explain that traditional assays in follow-up are used to assess the effect on each of these individual phenotypes. We have also added starvation survival and early fecundity data for key strains, rather than relying on growth alone.

(3) The claim that irld genes are predominantly expressed in sensory neurons is made using a dataset that did not have the expression profiles for non-neuronal tissues. Since tissues such as intestine, muscles and hypodermis have important roles in dictating organism-level phenotypes, it is essential to know whether the irld genes are expressed in these tissues. The tissue-restricted role of irld genes in starvation resistance that is proposed in the study can be addressed if the cell type-specific expression pattern of irld genes during normal and starvation conditions is known.

The reviewer is absolutely correct that the expression analysis included before was incomplete. We have extended this analysis to also include the Cao et al. (2017) single-cell RNA-seq data, which reports on all major tissues. These results show that the *irld* genes are predominantly expressed in sensory neurons, but that they are also expressed at lower levels in other tissues that could affect starvation resistance. We have been careful to explicitly point out that they are expressed in additional tissues.

This and other reviewer comments motivated us to use CRISPR to knock a reporter gene into the *irld-39* and *irld-52* loci to generate endogenous reporter genes for expression analysis. However, we were not able to visualize expression. We went one step further and generated high-copy transgenic arrays with a promoter::reporter fusions for *irld-39* and *irld-52*, but again we were not able to detect expression. These results are disappointing, and the effort delayed submission, but we feel better knowing that at least we tried.

The reviewer also mentions *irld* expression in fed and starved conditions. We were able to look at this with our published whole-worm RNA-seq data, and we now show in Figure S6 that the *irld* genes are expressed at very low levels (barely or not detectable) and that about half of them are up-regulated during starvation. None of them are significantly down-regulated. We believe this observation is consistent with the effect of the *irld* genes examined on starvation resistance.

– Figure 3D: For this gene enrichment analysis, genes with the same symbol prefix were considered as part of the same gene family (line 362 in Methods). However, gene symbol prefixes in *C. elegans* are not always representative of gene function (e.g. let, eat, unc etc.). Hence, searching for enrichment of gene symbol prefixes might lead to misleading and incomplete results. For example, not all of the 283 genes with the 'nhr' gene symbol prefix (that the authors report in Figure 3D) belong to the NHR gene family. Many of these 'nhr' genes are pseudogenes (nhr-75, nhr-83, nhr-220 etc.) and many other genes with the 'nhr' symbol do not have the C4-zinc finger DNA binding domain that is a characteristic of members of the NHR gene family. Furthermore, not all members of the NHR gene family have 'nhr' gene symbol prefixes (e.g. daf-12, dpr-1, odr-7, unc-55 etc.). Instead of looking for enrichment of gene symbol prefixes, the authors should search for enrichment of specific protein domains (InterPro, Pfam etc.). This might reveal enrichment of genes belonging to other functional categories, in addition to the irld gene family identified here.

Thank you for this comment and thorough explanation. We have performed a new analysis using the Pfam database and removed the previous analysis as suggested. The receptor L domain (defining *irld* genes) was found to be enriched along with several other domains highlighted in Figure 3.

– Figures 3J, 3K, 4A, 4B: Measuring the body length of worms after 48 hr of recovery from starvation should not be the only parameter to quantify the starvation resistance of a particular genotype. The authors should also perform the standard starvation survival assays (similar to data shown in Figure 2E) for irld gene manipulations in N2 and MY2147 genetic backgrounds and also for the strains in which the interaction of irld genes with insulin signaling was investigated.

We now include starvation survival results for each of these strains to complement the starvation recovery analysis. We have also added early fecundity data for key strains.

– Line 152: That authors should clarify that irld gene expression is restricted to sensory neurons among neuron types. The single-cell RNA-seq dataset they have utilized here reports expression only among neuron classes, but not in non-neuronal tissues. Firstly, the authors should repeat this analysis with the unthresholded dataset from Taylor et al. and remake figures 4H and figure S6 using expression data from both neuronal and non-neuronal tissues. Since non-neuronal tissues such as the intestine, muscles and hypodermis likely have important roles in determining starvation resistance of the animals, it is crucial to look at the expression of irld genes in these non-neuronal tissues as well. Secondly, the single-cell RNA-seq strategy in Taylor et al. was primarily designed to identify gene expression in neurons. Hence, the gene expression data from non-neuronal tissues might have not detected medium or weak expression of genes in non-neuronal tissues. Since the authors make the claim that irld-39 and irld-52 function primarily in sensory neurons to affect starvation resistance (line 159), it would be prudent to make Crispr-based transcriptional GFP reporters for these two genes and validate whether their expression is indeed restricted to ASJ and ADL neurons, respectively. These reporters should also be used to demonstrate whether the expression of irld genes changes in these neurons during exposure to starvation. If no expression is detected in non-neuronal tissues, this would strengthen the argument that irld genes are expressed in an anatomically restricted manner only in specific sensory neurons, but they regulate starvation resistance at the organism level potentially via systemic signaling. This would signify presence of cell non-autonomous effects of irld genes, which can be investigated in future studies.

We agree that the analysis of single-cell RNA-seq data was not at all well-presented, and that we were remiss in not considering non-neuronal expression. We have incorporated analysis from Cao et al., 2017 to show that *irld* genes are typically expressed in sensory neurons, but they do indeed have expression in other cell types (Figure S7). We have also shown that *irld* genes are typically expressed at low levels in whole worms by re-analyzing our previously published data on fed and starved L1 larvae. *irld* genes are also up-regulated in starved L1s, which is now included (Figure S6). In addition to adding analysis from these two datasets, we spent several months generating new reporter strains but were unable to visualize expression, presumably because of low expression levels. Because the manuscript already includes a novel application of an innovative sequencing approach, identification of multiple specific variants, and determining causality of those variants, we view it as outside the scope of the manuscript to decisively determine site of action, though it is a very interesting question.

Reviewer #3 (Recommendations for the authors):The authors make two major findings. First, they adapt MIP-seq to *C. elegans*, identifying primers that can identify individual strains and quantitate their relative proportion in group competition experiments. Second, they use this technique to map loci responsible for natural variation in starvation response, identifying irld family member genes that influence survival to starvation.The development of MIP-seq is a major achievement. Other labs can use the primers that they develop to perform similar experiments, competing wild strains against each other in their assay of interest. Because the wild strains are already sequenced, GWAS can be performed without the cost of any sequencing, using available software that is provided by the authors. One potential issue with the adaptability of this approach is the potential for outcrossing during the competition phase. While this is not an issue for the starvation mapping they perform here, it will need to be addressed to make this technique generalizable. However, enthusiasm for this approach remains high; competition in large group settings provides a much better handle on fitness than more common assays.Besides mapping and validating the loci that are responsible for variation in starvation response, they also identify causal mutations in irld genes. Using CRISPR-Cas9, they demonstrate a role for two natural mutations in starvation response and also implicate two additional irld genes as well. Use of CRISPR-Cas9 to specifically edit the genome are the gold standard for demonstrating a causal role for specific mutations. irld genes are homologous to insulin/EGF receptor proteins and this work implicates insulin signaling in starvation response. Additionally, they use classical genetics to implicate insulin signaling using epistasis experiments with the FOXO DAX-16 transcription factor. In general, starvation is poorly understood in humans and other species. This provides important evidence that insulin might be involved.

Thank you for highlighting the strengths of our study.

1. It is interesting that these genes are primarily expressed in sensory neurons. It is probably useful to use rescue experiments to show that this is the case.

We agree that it is very interesting that the *irld* genes, including *irld-39* and *irld-52*, are primarily expressed in sensory neurons. In addition to Taylor et al. (2021), we have added analysis of two more previously published datasets (Webster et al., 2018 and Cao et al., 2017). In particular, these datasets show that *irld* genes are expressed at low levels in whole worms, are up-regulated in starvation, are primarily expressed in sensory neurons, and are expressed in a few other cell types as well. The Cao data provide single-cell resolution across all major tissue types, while the Taylor data provide very high resolution of different neuronal cell types in particular. We did make reporter strains (CRISPR knock-ins as well as high-copy transgenic promoter fusions), but we were unable to visualize expression of the *irld* genes, unfortunately. We also note that expression data alone does not confirm that the *irld* genes act in these tissues, and we agree that genetic analysis of the site of action is desirable. However, we have been working on this project for several years, having started with the development of the MIP-seq assay and associated liquid-culture protocols and gone all the way through GWA, generation and analysis of NILs, generation and analysis of CRISPR edits, and genetic analysis in N2. At this point the first author has been out of the lab for a year, other personnel have moved on, and we believe the best thing for the field would be get this work published.

2. Since the strains exists, some experiments on the CRISPR/Cas9 allelic-replacement strains in normal well-fed conditions would be useful. Do these strains have the same lifespan, store the same amount of fat, and eat the same amount of food during non-starvation conditions that could help explain why they survive longer?

These are interesting questions. Some experiments that we did as controls for starvation resistance do lend insight into how the alleles behave under well-fed conditions. *irld-39(duk1); irld-52(duk17)* worms exhibit a significant interaction between strain and starvation duration for recovery growth and early fecundity. While *irld-39(duk1); irld-52(duk17)* worms recover quicker and produce more progeny than N2 following a longer starvation period, they recover slightly more slowly and produce fewer progeny than N2 following only brief starvation (a duration typically used to synchronize worms). This suggests a trade-off between starvation resistance and growth rate under fed conditions. Further supporting this, we see a modest trade-off across all strains in the MIP-seq experiment between their starvation resistance trait value and their ability to recover after just a single day of starvation (Figure S4). However, we have not looked at lifespan, feeding behavior, fat storage, dauer formation, etc, and we believe it would fall outside the scope of this work. For variation in feeding or fat accumulation to affect starvation resistance would require intergenerational effects, since worms starved during L1 arrest have never been fed. Intergenerational effects on starvation resistance are possible, as we have shown (Hibshman
*et al.* 2016; Jordan
*et al.* 2019), but in this case we propose a more proximate mechanism involving modification of *daf-16/FoxO* activity, which we know directly regulates starvation resistance.

3. There is little discussion about the generalizability of this work to other species. What is known or thought about variation in insulin pathways and starvation? Are there specific example in other species? How does this affect how we think about starvation and natural variation in starvation in humans and other species?

Thank you for this suggestion. We have significantly expanded the Introduction and Discussion sections. We cite papers on the importance of insulin/IGF signaling to starvation resistance in *C. elegans* and the importance of insulin signaling in starvation resistance and metabolic syndrome in mammals. Though the *irld* gene family is not conserved outside of the genus, we suggest that it provides an example of how expansion or contraction of a gene family can influence adaptation by modifying the activity of an essential, conserved signaling pathway. We also draw a parallel between *irld* gene function and IGF-binding proteins, and we specifically suggest that variation in that family may influence phenotypic variation in humans and other vertebrates.

References:

Baugh, L. R., and P. J. Hu, 2020 Starvation Responses Throughout the *Caenorhabditis elegans* Life Cycle. Genetics 216**:** 837-878.

Chen, J., L. Y. Tang, M. E. Powell, J. M. Jordan and L. R. Baugh, 2022 Genetic analysis of daf-18/PTEN missense mutants for starvation resistance and developmental regulation during *Caenorhabditis elegans* L1 arrest. G3 (Bethesda).

Dlakic, M., 2002 A new family of putative insulin receptor-like proteins in *C. elegans*. Curr Biol 12**:** R155-157.

Hibshman, J. D., A. Hung and L. R. Baugh, 2016 Maternal Diet and Insulin-Like Signaling Control Intergenerational Plasticity of Progeny Size and Starvation Resistance. PLoS Genet 12**:** e1006396.

Jobson, M. A., J. M. Jordan, M. A. Sandrof, J. D. Hibshman, A. L. Lennox *et al.*, 2015 Transgenerational Effects of Early Life Starvation on Growth, Reproduction, and Stress Resistance in *Caenorhabditis elegans*. Genetics 201**:** 201-212.

Jordan, J. M., J. D. Hibshman, A. K. Webster, R. E. W. Kaplan, A. Leinroth *et al.*, 2019 Insulin/IGF Signaling and Vitellogenin Provisioning Mediate Intergenerational Adaptation to Nutrient Stress. Curr Biol 29**:** 2380-2388 e2385.

Martinez, B. A., P. Reis Rodrigues, R. M. Nunez Medina, P. Mondal, N. J. Harrison *et al.*, 2020 An alternatively spliced, non-signaling insulin receptor modulates insulin sensitivity via insulin peptide sequestration in *C. elegans*. *ELife* 9.

Mata-Cabana, A., L. Gomez-Delgado, F. J. Romero-Exposito, M. J. Rodriguez-Palero, M. Artal-Sanz *et al.*, 2020 Social Chemical Communication Determines Recovery From L1 Arrest via DAF-16 Activation. Front Cell Dev Biol 8**:** 588686.

Webster, A. K., A. Hung, B. T. Moore, R. Guzman, J. M. Jordan *et al.*, 2019 Population Selection and Sequencing of *Caenorhabditis elegans* Wild Isolates Identifies a Region on Chromosome III Affecting Starvation Resistance. G3 (Bethesda) 9**:** 3477-3488.

Webster, A. K., J. M. Jordan, J. D. Hibshman, R. Chitrakar and L. R. Baugh, 2018 Transgenerational Effects of Extended Dauer Diapause on Starvation Survival and Gene Expression Plasticity in *Caenorhabditis elegans*. Genetics 210**:** 263-274.

[Editors’ note: what follows is the authors’ response to the second round of review.]

The manuscript has been improved but there are some remaining issues that need to be addressed, as outlined below:The reviewers have discussed their reviews with one another quite extensively. As you can see from their initial set of comments below, there still remains a substantial concern among all reviewers about whether and to what extent irld genes are involved in the starvation response. However, all reviewers have appreciated your creative new use of the MIP-Seq technology which we all expect to be quite impactful for further studies in *C. elegans*. We recommend that you (a) shorten the paper to a Report format, and (b) focus the paper on the technology aspect and provide the irld genes as an application. The present abstract of your paper organizationally already hints toward such a format, but this strategy should be implemented for the rest of the paper as well (including the Introduction). Also, as indicated by all of the reviewer's comments, please revise the manuscript editorially to consider alternative explanations of your data since, as you can see, the reviewers remain unconvinced that irld genes have been strongly implicated in this phenomenon.

Thank you for reviewing our revised manuscript and for providing guidance in how best to further revise it. We have substantially shortened the main text of our manuscript to better fit the Short Report format. We now emphasize the MIP-seq methodology, which was recognized as broadly useful to the community. In addition, we have added caveats and alternative interpretations throughout. In particular, we agree with the reviewers that while *irld* genes play a role in starvation recovery in the N2 and MY2147 backgrounds, they do not explain the phenotypic variation captured by the NILs, and it is likely that other genes within the QTL are involved. Below, we highlight the specific changes we made to the manuscript in response to reviewer comments. Our responses are in blue.

Reviewer #1:In the revised format, the manuscript is coherent and concise. This study has used MIP-seq to identify that genetic variation in the irld gene family determines some aspects of starvation resistance in wild worm isolates. However, the manuscript lacks any evidence regarding the mechanism and site of action of irld genes except the finding that one of the three phenotypes is DAF-16-dependent. Given the limited depth and breadth of this study, it is more suitable for the Short Reports format rather than the Research Article format.

Thank you for recognizing the strengths of our revised manuscript, especially with regard to MIP-seq and identification of the *irld* gene family as impacting aspects of starvation resistance. We recognize that we have not provided experimental evidence in support of mechanism or site of action, and we appreciate the suggestion to instead publish the manuscript as a Short Report.

Regarding the response to major comment #2 (Reviewer 2), the authors have not measured the starvation resistance of the irld-39; irld-52 double mutant in the MY2147 background. They quantified these phenotypes only in the N2 background, where the effects are either modest or not significant. Since single mutant manipulations produce much stronger phenotypes in the MY2147 background compared to N2 (Figure 3K), it is likely that the double mutant might display a robust increase in starvation resistance in the stress-sensitive MY2147 background. This experiment, though not essential, will greatly increase the impact of the study and will indicate that simultaneous manipulation of only a handful of irld genes can completely ameliorate the high stress-sensitivity in a wild strain.

Thank you for making this suggestion. It is an interesting experiment, but we suspect that the trait is sufficiently polygenic such that even this double mutant would not account for the phenotypic variation seen among the NILs. Since this experiment is not essential, we chose to focus on the guidance provided in the decision letter for how to revise the manuscript.

Reviewer #2:I appreciate the efforts that the authors have undertaken to address my critique of the initial submission, and I also commend them for their transparency about their results. However, I remain unconvinced about two of the authors' claims: that the irld variants they focus on account for the differences in starvation resistance observed in wild strains, and that irld-39/52 act through DAF-16 to modulate starvation resistance.

Thank you for recognizing the effort we put into revising the manuscript and for commending us for transparency. It was never our intention to imply that the *irld* variants investigated account for the full extent of the differences seen in wild strains or NILs, and we regret it if we inadvertently implied this. We now make it clear that we do not believe they do so, but we do think that they likely contribute to phenotypic variation (with caveats, discussed below). As for acting through DAF-16, we present the results of a standard epistasis analysis which suggests that the effect of the *irld-39; irld-52* double mutant depends on *daf-16.* We recognize that additional lines of evidence to further support this conclusion would be desirable, and that is why we included DAF-16 nuclear localization. While this result is subject to interpretation, it is nonetheless consistent with altered DAF-16 activity, and we clearly indicate its limitations (see below).

1. Figures 3I/J: this data shows that the irld edited strains do not confer improved starvation survival on the sensitive MY2147 background. While the authors offer the interpretation that "…the variants primarily affect starvation recovery," another more parsimonious explanation would be that these variants are not the key functional variants within the QTLs identified using MIP-seq.

We completely agree that these variants do not fully account for the phenotypic variation associated with the QTLs. The effect sizes are indicated in microns on each figure panel, allowing the reader to directly compare results (eg, NILs and *irld* variants). We also believe it should be made clear that only certain aspects of starvation resistance are affected. We have revised our conclusion accordingly (lines 211-214): "These results show that multiple types of variants in different *irld* family members reduce the effect of extended L1 starvation on recovery, suggesting four individual genes from this family affect this aspect of starvation resistance in wild strains. Notably, none of the engineered variants affected the trait to a similar extent as the NILs, suggesting that other variants within each QTL also affect the trait."

We also make this point in the Discussion (lines 267-269): "However, *irld* variants investigated each had relatively weak phenotypic effects compared to the NILs, suggesting they do not fully account for natural variation in the trait associated with the QTLs. This implies other variants (Supplementary File 2), possibly of larger effect, also contribute to phenotypic variation." And we cite Supplementary File 2 where additional candidates are catalogued.

2. The authors mention that "…the irld double mutant did not cause statistically significant changes in the expression of individual genes." Their explanation for this observation is that "We believe there must be differences in gene expression, but that they are relatively small and in specific tissues, thus obscured by analyzing whole worms. We also believe this is a testament to the sensitivity of our phenotypic assays." To me, the most parsimonious explanation for the observation that DAF-16 target gene expression is not influenced by irld-39/52 mutation is that the small increase in DAF-16 nuclear localization observed is not functionally significant. Moreover, while their phenotypic assays may be sensitive, the other more straightforward (IMO) explanation is that their phenotypic assays are capturing effects of other variants within the identified QTLs that are distinct from the irld variants that the authors have chosen to focus on.

We certainly agree that there are likely other important variants affecting the trait, as we now make clear, but we chose to focus on the *irld* genes because of their novelty, potential influence on insulin/IGF signaling, and impact on starvation recovery. As for DAF-16 localization, we accept that we show a relatively small effect. But the phenotypic effects of the *irld* genes are also of small effect, so we don't think this is surprising. Though difficult to study, we believe that small effects are important, especially in trying to understanding the genetic basis of polygenic traits. From a probabilistic perspective, any additional time DAF-16 spends in the nucleus provides additional opportunities for it to interact with DNA and affect transcription, with cumulative impact, and there is no reason to think that small differences in gene expression don't have phenotypic consequences, even if our assays don't have the power to detect those differences in expression. It is also important to note that we assayed DAF-16 localization in the intestine, which is typical given the relatively large size of these cells. The intestine is an important site of *daf-16* action for starvation resistance, but it is not the only site of action (eg, the nervous system is also important), and it is unclear if it is the most salient site in this context. We now state these limitations of the DAF-16 localization results (lines 238-239): "However, this is a relatively modest difference in nuclear localization, and it is unclear where in the animal DAF-16 activity is most relevant in this context." Despite the limitations of this assay, we believe it is valuable in that it complements the results of genetic epistasis.

3. The QTLs identified by the authors range from >600kb to >2.2Mb by my estimate. How many polymorphisms lie within these intervals? I understand the interest in the irld gene family, but the data do not convince me that the irld variants in question are the key functional variants within these intervals.

Yes, these are excellent points that we now make clearly in the manuscript. It was not our intention to claim that the *irld* genes are the key functional variants in the QTL, but rather that they are examples of functional variants within these intervals that affect this polygenic trait, and they are of broad interest because they are almost completely uncharacterized and potentially modify insulin/IGF signaling. To ensure that we do not leave readers with the wrong impression, we have revised the text in multiple places to explicitly indicate that other variants are likely involved. All variants are listed in Supplemental Data 2, and enrichment analyses in Figure 3 clearly indicate large numbers of genes in other gene families that contain variants. We now state on lines 164-165 that "These QTL are relatively large, ranging from 0.7 to 2.2 Mb, and include many candidate variants (Supplementary File 2) across 867 genes."

Reviewer #3:The authors utilize a standard starvation assay to study natural variation in starvation response among wild strains of *C. elegans*. Taking advantage of the CeNDR database, the authors compete ~100 wild strains against each other and quantify the changes in population using MIP-seq. The authors quite convincingly show that differences in survival occur among the wild strains and use GWAS to non-biasedly identify regions of the genome that are associated with differences in survival in this paradigm (QTLs). Technically this is quite challenging, and the identification of the QTLs was verified nicely using near isogenic lines.

Thank you for recognizing the strengths of our manuscript.

Throughout the paper, the authors describe the differences in survival in these conditions as starvation resistance. However, additional factors could be at play, such as differential susceptibility to toxins or pheromones that could build up during the multiday experiment.

We appreciate the concern, recognizing that factors in addition to starvation may affect survival in our assays. However, it is standard in the field to interpret differences in survival in these starvation conditions as differences in starvation resistance, and we assert that this is the simplest interpretation. Population density influences survival as well, and we carefully control density so that it is not a confounder. Temperature, maternal age, and maternal diet can also influence survival, and we carefully control each of these factors as well, as described in Materials and methods.

By analyzing the QTLs, the authors identified a family of genes, irlds, which were enriched within these regions. These genes are upregulated by starvation and have homology to insulin-type receptors. The authors propose that natural variation in these genes is important for natural variation in starvation response. Two demonstrate this, the authors identify two irld genes that carry likely loss of function deletions and use CRISPR/Cas9 to engineer these mutations into other genetic backgrounds. Additionally, the authors' engineer deletion alleles (that do not mimic segregating alleles) in two additional irld genes.Using these alleles, the authors convincingly show that irld genes play a role in this starvation assay, demonstrating differences in growth during and after exit from starvation. This result is likely to be exciting to researchers interested in starvation, as insulin signaling is an important genetic pathway in a large number of organisms.

Thank you for summarizing our findings. We appreciate your recognition that differences in growth and reproductive success after starvation reflect a role in the starvation response. We also agree that the starvation and aging fields should be excited to learn about the *irld* genes in this context.

While the authors also interpret their data to conclude that natural genetic variation in irld genes is important for natural variation in starvation resistance, I am less convinced by this data.

Thank you for raising these concerns. We appreciate this critique and now address each of these points in revision.

1) For the artificial deletion alleles of irld-57 and irld-11, functional differences in their protein activity or expression is not presented. How do the authors know that genetic variation among wild strains leads to functional differences in these proteins? Additionally, quantitative complementation (or a similar approach) was not performed to demonstrate that differences in their function exist between different wild strains.

The reviewer is correct that we have not shown functional differences in protein expression or activity for these genes in the relevant wild strains. Instead, this is presumed given a large number of variants in each gene predicted to disrupt protein function. We sidestepped this uncertainty by engineering deletion alleles, making their functional effect certain, but how well they serve as surrogates for the wild variants is unclear. We now make these points clear on lines 193-195: "Given several variants predicted to disrupt protein function in each, we believe *irld-11* and *irld-57* are null in the hyper-divergent context, though this has not been functionally demonstrated." And also on lines 202-205: "Since *irld-11* and *irld-57* contain so many candidate variants, we deleted these genes in MY2147 and N2, rendering them null at each locus (Figure 3H). Edits of *irld-39* and *irld-52* are more likely to approximate the effect of specific variants in the wild, because they are the exact variants present in starvation-resistant wild strains."

2) For the putative lof allele in irld-39, no data is shown to demonstrate that this affects IRLD-39 activity. This 5bp deletion is also close to the 5' end of a nearby gene – could the deletion be affecting the expression of this other gene?

The reviewer is correct again about these alternative possibilities. We agree that it is best to be clear about the limitations of the data presented. We now state on lines 179-182, "the variant is predicted to disrupt the start codon of the gene (Figure 3E, F, Supplementary File 2)*,* likely rendering *irld-39* a functional null in the starvation-resistant strain DL238. However, this was not functionally validated, and it is possible that this variant affects expression of the neighboring *irld* gene, *hpa-1.*"

3) The deletion of irld-52 is probably the most convincing, however, again, no evidence supporting its role as a loss of function allele is presented beyond sequence analysis such as isolation of cDNAs. Was this entire region sequenced to verify its existence and its predicted effect in wild strains (i.e. are there other genetic variants that might suppress the frameshift nature of this deletion?).

This point is also well taken, and we now state on lines 183-185 that *irld-52* "contains a variant associated with starvation resistance predicted to disrupt its fifth exon with a frameshift (Figure 3E, F), though this was not functionally validated and it is unclear if the variant causes a null mutation." We used the stringent version of the VCF file to identify variants, and we are not aware of any other variants that may suppress the frameshift.

We do not think it is sufficient to simply provide these caveats in the Results section, so we also incorporated them into the Discussion on lines 258-262: "we validated three QTL and showed four *irld* genes in these QTL impact starvation recovery. For *irld-11* and *irld-57*, we generated deletion mutants, which do not precisely match the variants present in wild strains. For *irld-39* and *irld-52,* the engineered alleles match starvation-resistant strains, but we have not confirmed their loss of function. Thus, our results suggest, but do not definitively demonstrate, that variation in *irld* genes affects starvation resistance in this species."

The authors also often switch back and forth between assays, which also makes it difficult to interpret the importance of the natural genetic variants in the overall differences in wild strains. Sometimes linear models are used to analyze strains (slope and intercept), sometimes size during starvation is used, and sometimes recovery is used.

We regret that it is complicated, but we believe that it is appropriate to break starvation resistance down into component phenotypes. We explain our rationale in the Introduction (more concisely in this version, but with references, including to a discussion of this issue in WormBook chapter), and we also touch on it in the Results section, so that the reader understands why multiple assays are reported. However, please note that we report starvation recovery (measured as length after 48 hours of recovery from brief or extended starvation), consistently throughout the manuscript, including in Figures 2F, 3K-L, 4B-D, and Figure 3—figure supplement 1C-D. In each case, the analysis is done using a linear mixed-effects model, which includes strain and days of starvation as fixed effects and biological replicate as a random effect. This was our primary assay, and upon revision we added in starvation survival data (we did not measure size during starvation) and fecundity data following starvation when relevant. It is of course difficult to compare these results to the MIP-seq assay (in which one trait value is the slope of a linear model), which is fundamentally different as a competition assay.

Because the NILs were not used as controls for many of these experiments, it is impossible to compare the effect of the individual mutations to the effect of the locus that has been mapped. Do these lof alleles represent the majority of the effect of this locus, or is this a minor effect and other much more important alleles remain to be found.

Thank you for bringing up this important point. You'll see in our response to Reviewer #2 above that we certainly do not believe that the *irld* variants identified represent the majority of the effect of these QTL, and this point is made explicit in the Results and Discussion.

We also originally wanted to directly compare variants and NILs to enable quantitative accounting of effect sizes. However, this is challenging, since not all of the associated variants are found in a given pair of strains, and a NIL of the appropriate background is not available for each comparison given known genetic incompatibilities we had to work around to generate NILs. Given these complications, our primary intent of genome editing was to identify variants that affect the trait, rather than accounting for the phenotypic variation associated with the QTL. We view the NILs and CRISPR-generated strains as complementary approaches; that is, the NILs show that the QTL underlie differences in starvation resistance between an appropriate pair of strains that differ for starvation resistance and for key variants within the QTL, while the CRISPR-generated strains show that specific genes within QTL with variants associated with starvation resistance impact the trait.

1. Either additional analysis to make the claim that the differences in survival is a starvation response or changes to the text to discuss alternative possibilities.

We believe this is in reference to the above comment about survival during starvation possibly reflecting the effect of environmental factors other than starvation itself, which we responded to above. In the interest of space, we do not agree that it is appropriate to discuss such alternative possibilities.

2. Further analysis to demonstrate that the natural 5bp deletions cause functional differences in IRLD protein

This is an important caveat that we now state explicitly on lines 179-182, "the variant is predicted to disrupt the start codon of the gene (Figure 3E, F, Supplementary File 2)*,* likely rendering *irld-39* a functional null in the starvation-resistant strain DL238. However, this was not functionally validated, and it is possible that this variant affects expression of the neighboring *irld* gene, *hpa-1.*" We chose to follow the guidance in the Editors' decision letter for revision rather than perform additional experiments to resolve this point.

3. Additional experiments or inclusion of existing data that allow the comparison of the effect size of the CRISP'ed strains to the appropriate NIL for comparison.

We agree that there are some instances in which a comparison to a NIL would have been possible, but there are a number of cases in which there is not an appropriate NIL for comparison, as stated above. The NILs and CRISPR edits serve complementary purposes and were not generated with the intent of being directly compared. However, we provide effect sizes on each figure panel, allowing the reader to clearly see that the effects of the *irld* variants are smaller than those of the NILs or wild strains. We also explicitly state in the Results and Discussion that the *irld* variants do not fully account for the phenotypic variation associated with the QTL or observed with the NILs.